# GENERATIVE MODELING OF REGULAR AND IRREGULAR TIME SERIES DATA VIA KOOPMAN VAES

**Ilan Naiman**
Ben-Gurion University, UC Berkeley
naimani@post.bgu.ac.il

**N. Benjamin Erichson**
LBNL and ICSI
erichson@lbl.gov

**Pu Ren**
LBNL
pren@lbl.gov

**Michael W. Mahoney**
ICSI, LBNL, and UC Berkeley
mmahoney@stat.berkeley.edu

**Omri Azencot**
Ben-Gurion University
azencot@cs.bgu.ac.il

## ABSTRACT

Generating realistic time series data is important for many engineering and scientific applications. Existing work tackles this problem using generative adversarial networks (GANs). However, GANs are unstable during training, and they can suffer from mode collapse. While variational autoencoders (VAEs) are known to be more robust to the these issues, they are (surprisingly) less considered for time series generation. In this work, we introduce Koopman VAE (KoVAE), a new generative framework that is based on a novel design for the model prior, and that can be optimized for either regular and irregular training data. Inspired by Koopman theory, we represent the latent conditional prior dynamics using a linear map. Our approach enhances generative modeling with two desired features: (i) incorporating domain knowledge can be achieved by leveraging spectral tools that prescribe constraints on the eigenvalues of the linear map; and (ii) studying the qualitative behavior and stability of the system can be performed using tools from dynamical systems theory. Our results show that KoVAE outperforms state-of-the-art GAN and VAE methods across several challenging synthetic and real-world time series generation benchmarks. Whether trained on regular or irregular data, KoVAE generates time series that improve both discriminative and predictive metrics. We also present visual evidence suggesting that KoVAE learns probability density functions that better approximate the empirical ground truth distribution.

## 1 INTRODUCTION

Generative modeling is an important problem in modern machine learning (Kingma & Welling, 2014; Goodfellow et al., 2014; Sohl-Dickstein et al., 2015), with a recent surge in interest due to results in natural language processing (Brown et al., 2020) and computer vision (Rombach et al., 2022; Ramesh et al., 2022). While image and text data have benefited from the recent development of generative models, time series (TS) data has received relatively little attention. This is in spite of the importance of generating TS data in various scientific and engineering domains, including seismology, climate studies, and energy analysis. Since these fields can face challenges in collecting sufficient data, e.g., due to high computational costs or limited sensor availability, high-quality generative models could be invaluable. However, generating TS data presents its own unique set of challenges. First, synthetic TS must preserve the related statistical distribution to fit into downstream forecasting, uncertainty quantification, and classification tasks. Second, advanced objectives such as supporting irregular sampling and integrating domain knowledge require models that respect the underlying dynamics (Che et al., 2018; Kidger et al., 2020; Jeon et al., 2022; Coletta et al., 2023).

Several existing state-of-the-art (SOTA) generative TS models are based on generative adversarial networks (GAN). For example, TimeGAN (Yoon et al., 2019) learns an embedding space where adversarial and supervised losses are optimized to mimic the data dynamics. Unfortunately, GANs are unstable during training and are prone to mode collapse, where the learned distribution is not sufficiently expressive (Goodfellow, 2016; Lucic et al., 2018; Saxena & Cao, 2021). In addition,

train sets with missing values or more generally, non-equispaced (irregularly-sampled) train sets, may be not straightforward to support in GAN architectures. For instance, Jeon et al. (2022) combine multiple different technologies to handle irregularly-sampled data, resulting in a complex system with many hyper-parameters. These challenges suggest that other generative paradigms should be also considered for generating time series information.

Surprisingly, variational autoencoders (VAEs) are not considered as strong baselines in generative time series benchmarks (Ang et al., 2023), although they suffer less from unstable training and mode collapse. In VAE, an approximate posterior is learned via a neural network to match a certain prior distribution that spans the data statistics. Recent methods including TimeVAE (Desai et al., 2021) and CR-VAE (Li et al., 2023) employ a variational viewpoint, however, their prior modeling is based on non-sequential standard normal priors. Thus, the learned posterior may struggle to properly represent the latent dynamics (Girin et al., 2021). This issue is further exacerbated when designing advanced sequential objectives. For instance, it is unclear how to incorporate domain knowledge about dynamical systems (e.g., stable dynamics, slow or fast converging or diverging dynamics) with unstructured Gaussian prior distributions. One promising direction to modeling latent dynamics is via *linear* approaches, that recently have been shown to improve performance and analysis (Zeng et al., 2023; Orvieto et al., 2023). More generally, this line of work aligns with theoretical and numerical tools from Koopman literature (Koopman, 1931; Rowley et al., 2009; Takeishi et al., 2017). Koopman theory offers a dual representation for autonomous dynamical systems via linear, albeit infinite-dimensional, operators. Harnessing this viewpoint facilitates autoencoder design, yielding finite-dimensional approximate Koopman operators that model the latent dynamics.

In this work, we propose Koopman VAE (KoVAE), a novel model that leverages linear latent Koopman dynamics within a VAE setup. KoVAE employs a prior stating that sequential latent data is governed by a *linear* dynamical system. Namely, the conditional prior distribution of the next latent variable, given the current variable, can be represented using a linear map. Under this assumption, we train an approximate posterior distribution that learns a nonlinear coordinate transformation of the inputs to a linear latent representation. Our approach offers two benefits: (i) it models the underlying dynamics and it respects the sequential nature of input data; and (ii) it seamlessly allows one to incorporate domain knowledge and study the qualitative behavior of the system by using spectral tools from dynamical systems theory. Moreover, we integrate into KoVAE a time-continuous module based on neural controlled differential equations (NCDE) (Kidger et al., 2020) to support irregularly-sampled time series information during training. The advantages of KoVAE are paramount as they facilitate the capturing of statistical and physical features of sequential information, and, in addition, imposing physics-constraints and analyzing the dynamics is straightforward.

**Contributions.** The main contributions of our work can be summarized as follows:

- We propose a new variational generative TS framework for regular and irregular data that is based on a **novel prior distribution** assuming an implicit linear latent Koopman representation. Our design and modeling yield a flexible, easy-to-code, and powerful generative TS model.

- We show that our approach facilitates **high-level capabilities** such as physics-constrained TS generation, by penalizing the eigenvalues of the approximated Koopman operator. We also perform stability analysis by inspecting the spectrum and its spread.

- We show improved state-of-the-art results in regular and irregular settings on several synthetic and real-world datasets, often surpassing strong baselines such as TimeGAN and GT-GAN by large margins. For instance, on the regular and irregular discriminative task we report a total mean relative improvement of **58%** and **49%** with respect to the second best approach, respectively.

## 2 RELATED WORK

Our work is related to generative models for TS as well as Koopman-based methods. Both areas enjoy increased interest in recent years, and thus, we focus our discussion on the most related work.

**Generative Models for Time Series.** Recurrent neural networks (RNNs) and their step-wise predictions have been used to generate sequences in Teacher forcing (Graves, 2013) and Professor forcing (Goyal et al., 2016) approaches. Autoregressive methods such as WaveNet (van den Oord et al., 2016) represent the predictive distribution of each audio sample by probabilistic conditioning on all previous samples, whereas TimeGCI (Jarrett et al., 2021) also incorporates GANs and optimizes a

transition policy where the reinforcement signal is provided by a global energy framework trained with contrastive estimation. Flow-based models for generating time series information utilize normalizing flows for an explicit likelihood distribution (Alaa et al., 2020). Long-range sequences were recently treated via state-space models (Zhou et al., 2023).

**GANs and VAEs.** GAN-based architectures have been shown to be effective for generating TS data. C-RNN-GAN (Mogren, 2016) directly applied GAN to sequential data, using LSTM networks for the generator and the discriminator. Recurrent Conditional GAN (Esteban et al., 2017) extends the latter work by dropping the dependence on the previous output while conditioning on additional inputs. TimeGAN (Yoon et al., 2019) is still one of the SOTA methods, and it jointly optimizes unsupervised and supervised loss terms to preserve the temporal dynamics of the training data during the generation process. COT-GAN (Xu et al., 2020) exploits optimal transport and temporal causal constraints to devise a new adversarial loss. Recently, GT-GAN (Jeon et al., 2022) was proposed as one of the few approaches that support irregular sampling, and it employs autoencoders, GANs, and Neural ODEs (Chen et al., 2018). While GAN techniques are more common in generative TS, there are also a few VAE-based approaches. A variational RNN was proposed in (Chung et al., 2015). TimeVAE (Desai et al., 2021) models the whole sequence by a global random variable with a normal Gaussian distribution for a prior, and they introduce trend and seasonality building blocks into the decoder. CR-VAE (Li et al., 2023) learns a Granger causal graph by predicting future data from past observations and using a multi-head decoder acting separately on latent variable coordinates. Additional related VAE works include (Rubanova et al., 2019; Li et al., 2020; Zhu et al., 2023).

**Koopman-based Approaches.** Koopman techniques have gained increasing interest over the past two decades, with applications ranging across analysis (Rowley et al., 2009; Schmid, 2010; Takeishi et al., 2017; Lusch et al., 2018; Azencot et al., 2019), optimization (Dogra & Redman, 2020; Redman et al., 2022), forecasting (Erichson et al., 2019; Azencot et al., 2020; Wang et al., 2023; Tayal et al., 2023), and disentanglement (Berman et al., 2023), among many others (Budišić et al., 2012; Brunton et al., 2021). Most related to our work are Koopman-based *probabilistic* models, which have received less attention in the literature. Deep variational Koopman models were introduced in Morton et al. (2019), allowing to sample from distributions over latent observables for prediction tasks. In Srinivasan & Takeishi (2020), the authors sample via Markov Chain Monte Carlo tools over transfer operators. A mean-field variational inference method with guaranteed stability was suggested in Pan & Duraisamy (2020) for the analysis and prediction of nonlinear dynamics. Finally, Han et al. (2022) designed a stochastic Koopman neural network for control that models the latent observables via a Gaussian distribution. To the best of our knowledge, our work is the first to combine VAEs and Koopman-based methods for generating regular and irregular TS information.

## 3 BACKGROUND

Below, we discuss background information essential to our method. In App. A and App. B, we also cover VAE and NCDE, used in our work to support irregularly-sampled sequential train sets.

**Koopman theory and practice.** The underlying theoretical justification for our work is related to dynamical systems and Koopman theory (Koopman, 1931). Let $\mathcal{M} \subset \mathbb{R}^m$ be a finite-dimensional domain and $\varphi : \mathcal{M} \to \mathcal{M}$ be a dynamical system defined by

$$x_{t+1} = \varphi(x_t) \, ,$$

where $x_t \in \mathcal{M}$, and $t \in \mathbb{N}$ is a discrete variable that represents time. Remarkably, under some mild conditions (Eisner et al., 2015), there exists an infinite-dimensional operator known as the *Koopman operator* $\mathcal{K}_\varphi$ that acts on observable functions $f : \mathcal{M} \to \mathbb{C} \subset \mathcal{F}$ and it fully characterizes the dynamics. The operator $\mathcal{K}_\varphi$ is given by

$$\mathcal{K}_\varphi f(x_t) = f \circ \varphi(x_t) \, ,$$

where $f \circ \varphi$ denotes composition of transformations. It can be shown that $\mathcal{K}_\varphi$ is linear, while $\varphi$ may be nonlinear. Further, if the eigendecomposition of the Koopman operator exists, the eigenvalues and eigenvectors bear dynamical semantics. Spectral analysis of Koopman operators is currently still being researched (Mezić, 2013; Arbabi & Mezic, 2017; Mezic, 2017; Das & Giannakis, 2019). For instance, eigenfunctions whose eigenvalues lie within the unit circle are related to global stability (Mauroy & Mezić, 2016), and to orbits of the system (Mauroy & Mezić, 2013; Azencot et al.,

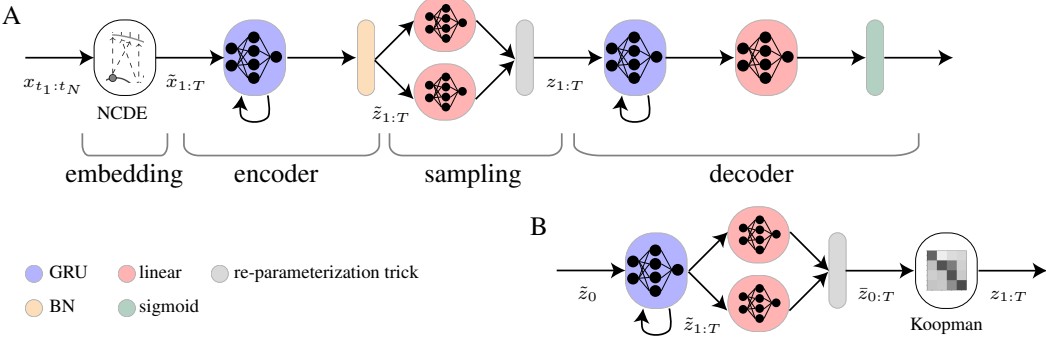

Figure 1: A) The posterior is composed of an embedding layer (NCDE), an encoder (GRU + BN), mean/variance computation and sampling (linear + repr. trick), and a decoder (GRU + linear + sigmoid). B) The prior consists of GRU, linear, a repr. trick layer, and our novel Koopman module.

2013; 2014). In practice, several methods have been devised to approximate the Koopman operator, of which, the dynamic mode decomposition (DMD) (Schmid, 2010) is perhaps the most well-known technique (Rowley et al., 2009). In this context, our work employs a learnable module akin to DMD in the model prior of our KoVAE as detailed below. In general, our approach can be viewed as treating the input data as states $x_{1:T}$, whereas the learnable linear dynamics we compute in KoVAE are parametrized by the latent observable functional space $\mathcal{F}$.

**Sequential VAEs.** We denote by $x_{1:T}$ a TS input $x_{1:T} = x_1, \ldots, x_T$, where $x_t \in \mathbb{R}^d$ for all $t$. Below, we focus on a general approach (Girin et al., 2021), where the joint distribution is given by

$$p(x_{1:T}, z_{1:T}) = p(z_{1:T})p(x_{1:T}|z_{1:T}) = \prod_{t=1}^{T} p(z_t|z_{<t}) \cdot \prod_{t=1}^{T} p(x_t|z_t) \,, \tag{1}$$

where $z_t$ is the prior latent code associated with $x_t$, it depends on all past variables $z_{<t}$, and given $z_t$, one can recover $x_t$ via a neural decoder $p(x_t|z_t)$. The approximate posterior is modeled by

$$q(z_{1:T}|x_{1:T}) = \prod_{t=1}^{T} q(z_t|z_{<t}, x_{\leq t}) \,, \tag{2}$$

namely, $z_t$ is the posterior latent code, it is generated using the learned encoder $q(z_t|z_{<t}, x_{\leq t})$, and the dynamic VAE loss is composed of reconstruction and regularization terms and it reads

$$\mathcal{L}_{\text{VAE}} = \mathbb{E}_{z_{1:T} \sim q}[\log p(x_{1:T}|z_{1:T})] - \text{KL}[q(z_{1:T}|x_{1:T}) \,\|\, p(z_{1:T})] \,. \tag{3}$$

## 4 KOOPMAN VARIATIONAL AUTOENCODERS (KOVAE)

To describe our generative model, we detail its sequential posterior in Sec. 4.1, its novel linear Koopman-based prior in Sec. 4.2, and its support for incorporating domain knowledge in Sec. 4.3.

### 4.1 A VARIATIONAL SEQUENTIAL POSTERIOR

To realize Eq. 2, our approximate posterior (Fig. 1A) is composed of embedding, encoder, sampling, and decoder modules. Given the irregular TS $x_{t_1:t_N}$, an NCDE embedding layer extracts a regularly-sampled TS $\tilde{x}_{1:T}$, followed by an encoder module consisting of a gated recurrent unit (GRU) layer (Cho et al., 2014) and batch normalization (BN) layer that learn the dynamics and output the latent representation $\tilde{z}_{1:T}$. Then, two linear layers produce the posterior mean $\mu(\tilde{z}_t)$ and variance $\sigma^2(\tilde{z}_t)$ for every $t$, allowing to employ the re-parameterization trick to generate the posterior series $z_{1:T}$ via

$$z_t \sim \mathcal{N}(\mu(\tilde{z}_t), \sigma^2(\tilde{z}_t)) \,. \tag{4}$$

We feed the sampled code to the decoder that includes another GRU, a linear layer, and a sigmoid activation function. Note that the embedding NCDE layer is not used in the regular setting, i.e., when the input TS $x_{1:T}$ is regularly sampled.

## 4.2 A NOVEL LINEAR VARIATIONAL SEQUENTIAL PRIOR

In general, one of the fundamental tasks in variational autoencoders' design and modeling is the choice of prior distribution. For instance, static VAEs define it to be a normal Gaussian distribution, i.e., $p(z) = \mathcal{N}(0, I)$, among other choices, e.g., (Huszár, 2017). In the sequential setting, a common choice for the prior is a sequence of Gaussians (Chung et al., 2015) given by

$$p(z_t|z_{<t}) = \mathcal{N}(\mu(z_{<t}; \theta), \sigma^2(z_{<t}; \theta)) , \qquad (5)$$

where the mean and variance are learned with a NN, and $z_{1:T}$ represents the latent sequence. Now, the question arises whether one can model the prior of generative TS models $p(z_{1:T}) = \prod_{t=1}^{T} p(z_t|z_{<t})$ in Eq. 1 so that it associates better with the underlying dynamics. In what follows, we propose a novel inductive bias leading to a new VAE modeling paradigm.

**Prior modeling.** In this work, instead of considering a nonlinear sequence of Gaussian as in Eq. 5, we assume that there exists a learnable nonlinear coordinate transformation mapping inputs to a *linear* latent space. That is, the inputs $x_{t_1:t_N}$ are transformed to $z_{1:T}$ whose dynamics are governed by a linear sequence of Gaussians, see inset for an illustration. Formally, the forward propagation in time of $z_t$ is governed by a matrix, i.e.,

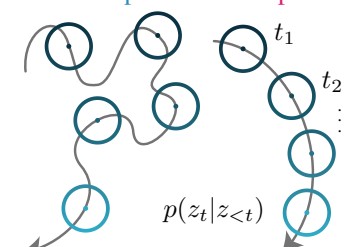

$$z_t := \mathbb{E}_{A \sim \mathcal{A}} [A z_{t-1}] , \qquad (6)$$

for every $t$, where $A \in \mathbb{R}^{k \times k}$ is sampled from a space of linear operators $\mathcal{A}$. In practice, we implement the prior (Fig. 1B) and produce $z_{1:T}$ using a gated recurrent unit (GRU) layer, a sampling component, and a Koopman module (Takeishi et al., 2017). Given $\tilde{z}_0 := \vec{0}$, the GRU yields $\tilde{z}_{1:T}$ that is fed to two linear layers which produce the mean $\mu(\tilde{z}_t)$ and variance $\sigma^2(\tilde{z}_t)$ for every t, allowing to sample $\bar{z}_t$ from $\tilde{z}_{1:T}$ via Eq. 5. To compute $A$, we construct two matrices $\bar{Z}_0, \bar{Z}$ with $\bar{z}_{0:T-1}$ and $\bar{z}_{1:T}$ in their columns, respectively, where $\bar{z}_0 \sim \mathcal{N}(\mu(0; \theta), \sigma^2(0; \theta))$. Then, we solve the linear system for the best $A$ such that $A\bar{Z}_0 = \bar{Z}$, similar to DMD (Schmid, 2010). Finally, $z_t$ is defined to be $z_t := A\bar{z}_{t-1}$.

**Training objective.** Effectively, it may be that $A$ induces some error, i.e., $z_t \neq \bar{z}_t$ for some $t \in [1, \ldots, T]$. Thus, we introduce an additional predictive loss term that promotes linearity in the prior latent variables by matching between $\bar{z}_{1:T}$ and $z_{1:T}$, namely,

$$\mathcal{L}_{\text{pred}}(z_{1:T}, \bar{z}_{1:T}) = \mathbb{E}_{\bar{z}_{1:T} \sim p}[\log p(z_{1:T}|\bar{z}_{1:T})] . \qquad (7)$$

Combining the penalties from Sec. 3 and the loss Eq. 7, we arrive at the following training objective which includes a reconstruction term, a prediction term, and a regularization term,

$$\mathcal{L} = \mathbb{E}_{z_{1:T} \sim q}[\log p(x_{t_1:t_N}|z_{1:T})] + \alpha \mathcal{L}_{\text{pred}}(z_{1:T}, \bar{z}_{1:T}) - \beta \text{KL}[q(z_{1:T}|x_{t_1:t_N}) \| p(z_{1:T})] , \qquad (8)$$

where $\alpha, \beta \in \mathbb{R}^+$ are user weights, similar to (Higgins et al., 2016). In App. C, we provide technical details on $\mathcal{L}$, and in App. D, we show that our objective is a penalized evidence lower bound (ELBO).

## 4.3 PHYSICS-CONSTRAINED GENERATION AND ANALYSIS

The matrix $A$ computed in Sec. 4.2 encodes the latent dynamics in a linear form. Thus, it can be constrained and evaluated using spectral tools from dynamical systems theory (Strogatz, 2018). For instance, the eigenvalues $\lambda_j \in \mathbb{C}, j = 1, \ldots, k$, are associated with growth ($|\lambda_j| > 1$) and decay ($|\lambda_j| < 1$), whereas the eigenvectors $\phi_j \in \mathbb{C}^k$ encode the dominant modes. Our constrained generation and interpretability results are based on the relation between $A$ and dynamical systems.

Often, prior knowledge of the problem can be utilized in the generation of time series information, see e.g., the example in Sec. 5. Our framework allows us to directly incorporate such knowledge by constraining the eigenvalues of the system, as was recently proposed in (Berman et al., 2023). Specifically, we denote by $c_1, \ldots, c_r \subset \mathbb{C}$ with $r \leq k$ several known constant values, and we define the following penalty we add to the optimization that yields $A$ whose eigenvalues are denoted by $\lambda_j$,

$$\mathcal{L}_{\text{eig}} = \sum_{j=1}^{r} ||\lambda_j| - c_j|^2 , \qquad (9)$$

where $|\lambda_j|$ is the modulus of a complex number. The loss term 9 can be used during training or inference, or both. For instance, stable dynamical systems as we consider in Sec. 5.2, have Koopman operators with eigenvalues on the unit circle, i.e., $|\lambda_j| = 1$. Thus, when constraining the dynamics, we can fix some $c_1, \ldots, c_r = 1$, and leave the rest to be unconstrained. In addition to constraining the spectrum, we can also analyze the spectrum to study the stability behavior of the system, similarly to (Erichson et al., 2019; Naiman & Azencot, 2023), see also Sec. 5.2.

## 5 EXPERIMENTS

In this section, we detail the results of our extensive experiments. Details related to datasets and baselines are provided in App. E. Our code is available at GitHub.

### 5.1 GENERATIVE TIME SERIES RESULTS

In what follows, we demonstrate our model's generation capabilities empirically, and we evaluate our method both quantitatively and qualitatively in the tests we describe below.

**Quantitative evaluation.** Yoon et al. (2019) suggested two tasks for evaluating generative time series models: the discriminative task and the predictive task. In the *discriminative test*, we measure the similarity between real and artificial samples. First, we generate a collection of artificial samples $\{y_{1:T}^j\}_{j=1}^N$ and we label them as 'fake'. Second, we train a classifier to discriminate between the fake and the real dataset $\{x_{1:T}^j\}_{j=1}^N$. Finally, we report the value $|\frac{1}{2} - acc|$, where acc is the accuracy of the discriminator on a held-out set. Thus, a lower discriminative score implies better generated data as it fools the discriminator. The *predictive task* is based on the "train on synthetic, test on real" protocol. We we train a predictor on the generated artificial samples. Then, the predictor is evaluated on the real data. A lower mean absolute error (MAE) indicates improved predictions.

We consider the discriminative and predictive tasks in two evaluation cases: *regular*, where we use the entire dataset; and *irregular*, where we omit a portion of the dataset. Specifically, in the irregular setting, we follow Jeon et al. (2022), and we randomly omit $30\%, 50\%$, and $70\%$ of the observations. We show in Tab. 1 and Tab. 2 the discriminative and predictive scores, respectively. Tab. 1 lists at the bottom the relative improvement (RI) of the best score $s_1$ with respect to the second best score $s_2$, i.e., $|s_2 - s_1|/s_2$. Tab. 2 details the predictive scores obtained when the real data is used in the task (original). Both tables highlight the best scores. The results show that our approach (KoVAE) attains the best scores on both tasks. In particular, KoVAE demonstrates significant improvements on the discriminative tasks, yielding a remarkable average RI of **58%**. In Tab. 3, we detail the discriminative and predictive scores for the irregular setting. Similarly to the regular case, KoVAE presents the best measures, except for predictive Stocks $50\%$ and discriminative Energy $70\%$. Still, our average RIs are significant: **48%**, **55%**, and **43%** for the three different irregular settings.

**Qualitative evaluation.** Next, we use qualitative metrics to examine the similarity of the generated sequences to the real data. We consider two visualization techniques: (i) we project the real and

Table 1: Regular TS, discriminative task

| Method | Sines | Stocks | Energy | MuJoCo |
|---|---|---|---|---|
| KoVAE (Ours) | **0.005** | **0.009** | **0.143** | **0.076** |
| GT-GAN | 0.012 | 0.077 | 0.221 | 0.245 |
| TimeVAE | 0.016 | 0.036 | 0.323 | 0.224 |
| TimeGAN | 0.011 | 0.102 | 0.236 | 0.409 |
| CR-VAE | 0.342 | 0.320 | 0.475 | 0.464 |
| RCGAN | 0.022 | 0.196 | 0.336 | 0.436 |
| C-RNN-GAN | 0.229 | 0.399 | 0.499 | 0.412 |
| T-Forcing | 0.495 | 0.226 | 0.483 | 0.499 |
| P-Forcing | 0.430 | 0.257 | 0.412 | 0.500 |
| WaveNet | 0.158 | 0.232 | 0.397 | 0.385 |
| WaveGAN | 0.277 | 0.217 | 0.363 | 0.357 |
| RI | **54.54%** | **75.00%** | **35.29%** | **66.07%** |

Table 2: Regular TS, predictive task

| Method | Sines | Stocks | Energy | MuJoCo |
|---|---|---|---|---|
| KoVAE (Ours) | **0.093** | **0.037** | **0.251** | **0.038** |
| GT-GAN | 0.097 | 0.040 | 0.312 | 0.055 |
| TimeVAE | **0.093** | 0.037 | 0.254 | 0.039 |
| TimeGAN | 0.093 | 0.038 | 0.273 | 0.082 |
| CR-VAE | 0.143 | 0.076 | 0.277 | 0.050 |
| RCGAN | 0.097 | 0.040 | 0.292 | 0.081 |
| C-RNN-GAN | 0.127 | 0.038 | 0.483 | 0.055 |
| T-Forcing | 0.150 | 0.038 | 0.315 | 0.142 |
| P-Forcing | 0.116 | 0.043 | 0.303 | 0.102 |
| WaveNet | 0.117 | 0.042 | 0.311 | 0.333 |
| WaveGAN | 0.134 | 0.041 | 0.307 | 0.324 |
| Original | 0.094 | 0.036 | 0.250 | 0.031 |

Table 3: Irregular time series (30%, 50% and 70% of observations are dropped)

| | | 30% | | | | 50% | | | | 70% | | |
|---|---|---|---|---|---|---|---|---|---|---|---|---|
| | | Sines | Stocks | Energy | MuJoCo | Sines | Stocks | Energy | MuJoCo | Sines | Stocks | Energy | MuJoCo |
| **Discriminative Score** | KoVAE (Ours) | **0.035** | **0.162** | **0.280** | **0.123** | **0.030** | **0.092** | **0.298** | **0.117** | **0.065** | **0.101** | 0.392 | **0.119** |
| | GT-GAN | 0.363 | 0.251 | 0.333 | 0.249 | 0.372 | 0.265 | 0.317 | 0.270 | 0.278 | 0.230 | **0.325** | 0.275 |
| | TimeGAN-$\triangle t$ | 0.494 | 0.463 | 0.448 | 0.471 | 0.496 | 0.487 | 0.479 | 0.483 | 0.500 | 0.488 | 0.496 | 0.494 |
| | RCGAN-$\triangle t$ | 0.499 | 0.436 | 0.500 | 0.500 | 0.406 | 0.478 | 0.500 | 0.500 | 0.433 | 0.381 | 0.500 | 0.500 |
| | C-RNN-GAN-$\triangle t$ | 0.500 | 0.500 | 0.500 | 0.500 | 0.500 | 0.500 | 0.500 | 0.500 | 0.500 | 0.500 | 0.500 | 0.500 |
| | T-Forcing -$\triangle t$ | 0.395 | 0.305 | 0.477 | 0.348 | 0.408 | 0.308 | 0.478 | 0.486 | 0.374 | 0.365 | 0.468 | 0.428 |
| | P-Forcing-$\triangle t$ | 0.344 | 0.341 | 0.500 | 0.493 | 0.428 | 0.388 | 0.498 | 0.491 | 0.288 | 0.317 | 0.500 | 0.498 |
| | TimeGAN-D | 0.496 | 0.411 | 0.479 | 0.463 | 0.500 | 0.477 | 0.473 | 0.500 | 0.498 | 0.485 | 0.500 | 0.492 |
| | RCGAN-D | 0.500 | 0.500 | 0.500 | 0.500 | 0.500 | 0.500 | 0.500 | 0.500 | 0.500 | 0.500 | 0.500 | 0.500 |
| | C-RNN-GAN-D | 0.500 | 0.500 | 0.500 | 0.500 | 0.500 | 0.500 | 0.500 | 0.500 | 0.500 | 0.500 | 0.500 | 0.500 |
| | T-Forcing-D | 0.408 | 0.409 | 0.347 | 0.494 | 0.430 | 0.407 | 0.376 | 0.498 | 0.436 | 0.404 | 0.336 | 0.493 |
| | P-Forcing-D | 0.500 | 0.480 | 0.491 | 0.500 | 0.499 | 0.500 | 0.500 | 0.500 | 0.500 | 0.449 | 0.494 | 0.499 |
| **Predictive Score** | KoVAE (Ours) | **0.074** | **0.019** | **0.049** | **0.043** | **0.072** | 0.019 | **0.049** | **0.042** | **0.076** | **0.012** | **0.052** | **0.044** |
| | GT-GAN | 0.099 | 0.021 | 0.066 | 0.048 | 0.101 | **0.018** | 0.064 | 0.056 | 0.088 | 0.020 | 0.076 | 0.051 |
| | TimeGAN-$\triangle t$ | 0.145 | 0.087 | 0.375 | 0.118 | 0.123 | 0.058 | 0.501 | 0.402 | 0.734 | 0.072 | 0.496 | 0.442 |
| | RCGAN-$\triangle t$ | 0.144 | 0.181 | 0.351 | 0.433 | 0.142 | 0.094 | 0.391 | 0.277 | 0.218 | 0.155 | 0.498 | 0.222 |
| | C-RNN-GAN-$\triangle t$ | 0.754 | 0.091 | 0.500 | 0.447 | 0.741 | 0.089 | 0.500 | 0.448 | 0.751 | 0.084 | 0.500 | 0.448 |
| | T-Forcing-$\triangle t$ | 0.116 | 0.070 | 0.251 | 0.056 | 0.379 | 0.075 | 0.251 | 0.069 | 0.113 | 0.070 | 0.251 | 0.053 |
| | P-Forcing-$\triangle t$ | 0.102 | 0.083 | 0.255 | 0.089 | 0.120 | 0.067 | 0.263 | 0.189 | 0.123 | 0.050 | 0.285 | 0.117 |
| | TimeGAN-D | 0.192 | 0.105 | 0.248 | 0.098 | 0.169 | 0.254 | 0.339 | 0.375 | 0.752 | 0.228 | 0.443 | 0.372 |
| | RCGAN-D | 0.388 | 0.523 | 0.409 | 0.361 | 0.519 | 0.333 | 0.250 | 0.314 | 0.404 | 0.441 | 0.349 | 0.420 |
| | C-RNN-GAN-D | 0.664 | 0.345 | 0.440 | 0.457 | 0.754 | 0.273 | 0.438 | 0.479 | 0.632 | 0.281 | 0.436 | 0.479 |
| | T-Forcing-D | 0.100 | 0.027 | 0.090 | 0.100 | 0.104 | 0.038 | 0.090 | 0.113 | 0.102 | 0.031 | 0.091 | 0.114 |
| | P-Forcing-D | 0.154 | 0.079 | 0.147 | 0.173 | 0.190 | 0.089 | 0.198 | 0.207 | 0.278 | 0.107 | 0.193 | 0.191 |
| | Original | 0.071 | 0.011 | 0.045 | 0.041 | 0.071 | 0.011 | 0.045 | 0.041 | 0.071 | 0.011 | 0.045 | 0.041 |

synthetic data into a two-dimensional space using t-SNE (Van der Maaten & Hinton, 2008); and (ii) we perform kernel density estimation by visualizing the probability density functions (PDF). In Fig. 2, we visualize the synthetic and real two-dimensional point clouds in the irregular $50\%$ setting for all datasets via t-SNE (top row), and additionally, we visualize their corresponding probability density functions (bottom row). Further, we also show the PDF of GT-GAN (Jeon et al., 2022) in dashed-black curves at the bottom row. Overall, our approach displays strong correspondences in

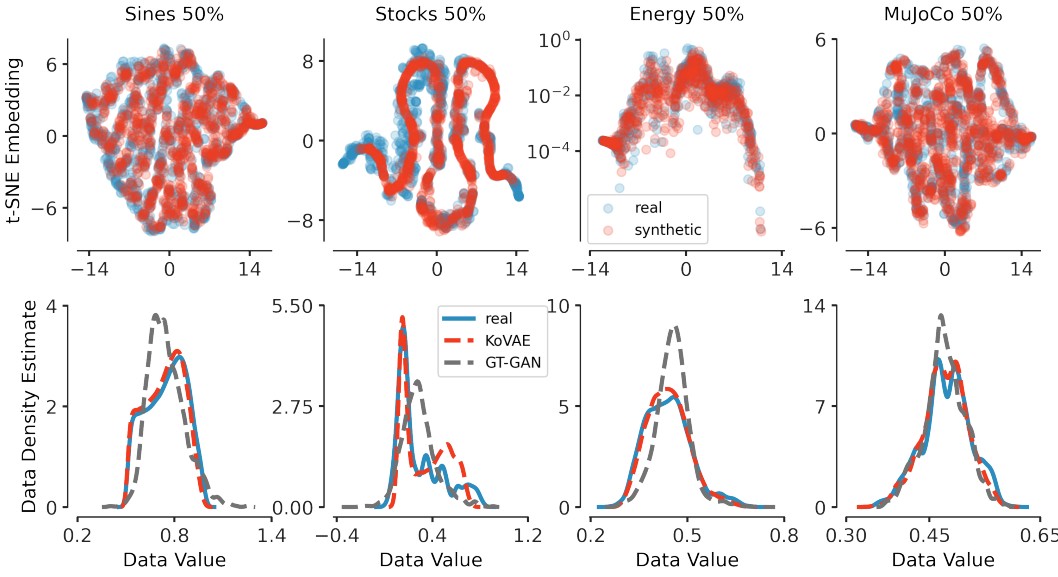

Figure 2: We qualitatively evaluate our approach with two-dimensional t-SNE plots of the synthetic and real data (top row). In addition, we show the probability density functions of the real data, and for KoVAE and GT-GAN synthetic distributions (bottom row).

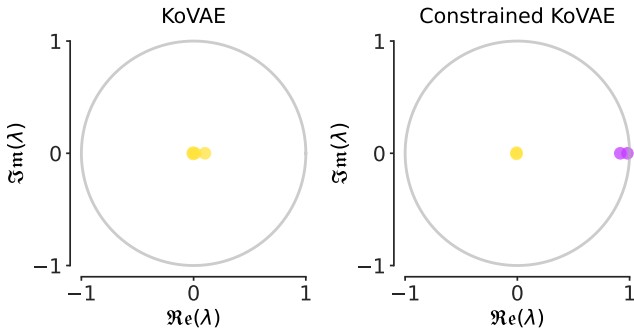

Figure 3: On the left, we show the spectrum of the approximate Koopman operator without constraints during the training. On the right, we show the spectrum of the approximate Koopman operator for the model that is trained with stability constraints. We can see that, indeed, the absolute value of two eigenvalues is approximately 1.

both visualizations, where in Fig. 2 (top row), we observe a high overlap between real and synthetic samples, and in Fig. 2 (bottom row), the PDFs show a similar trend and behavior. On Stocks $50\%$, we identify lower variability in KoVAE in comparison to the real data that present a larger variance. In App. H, we provide additional results for the regular and irregular $30\%$, and $70\%$ cases.

## 5.2 PHYSICS-CONSTRAINED GENERATION

In this section, we demonstrate how to incorporate prior knowledge of the problem when generating time series information. While we could study the Sines dataset, we opted for a more challenging test case, and thus, we focus on the *nonlinear pendulum* system. Let $l = 1$ and $g = 9.8$ denote the length and gravity, respectively. Then, we consider the following ordinary differential equation (ODE) that describes the evolution of the angular displacement from an equilibrium $\theta$,

$$\frac{\mathrm{d}^2\,\theta}{\mathrm{d}\,t^2} + \frac{g}{l}\sin\theta = 0\,, \quad \theta(0) = 0\,, \quad \dot{\theta}(0) = 0\,. \tag{10}$$

To generate $N$ different sequences, we uniformly sample $\theta^j(0) \sim \mathcal{U}(0.5, 2.7)$ for $j = 1, \dots, N$ over the time interval $t = [0, 17]$, where the time step is defined by $\Delta t = 0.1$. This results in a set $\{x^j_{1:T}\}^N_{j=1}$ with $T = 170$ and each time sample is two-dimensional, i.e., $x_t \in \mathbb{R}^2$. To simulate real-world noisy sensors, we incorporate additive Gaussian noise to each sample. Namely, we sample $\rho \sim \mathcal{N}(0, 1)$, and we define the train set samples via $\bar{x}_t := x_t + 0.08\,\rho$.

We evaluate the nonlinear pendulum on three different models: (i) a KoVAE with $\alpha = 0$; (ii) a KoVAE; and (iii) a KoVAE with an eigenvalue constraint as described in Sec. 4.3. Specifically, we train all models with a fixed latent size $\kappa$ of $z_t$ to be $\kappa = 4$. For the constrained version, we additionally add $\mathcal{L}_{\text{eig}}$:

$$\mathcal{L}_{\text{eig}} = ||\lambda_p| - c_p|^2 + ||\lambda_q| - c_q|^2\,, \tag{11}$$

where $c_p = 1, c_q = 1$ and $\lambda_p$ and $\lambda_q$ are the largest eigenvalues during training. We define this constraint since the nonlinear pendulum is a stable dynamical system that is governed by *two* modes. The rest of the eigenvalues, $\lambda_i$, where $i \neq p, q$ could be additionally constrained to be equal to zero. However, we observe that when we leave the rest of the eigenvalues to be unconstrained, we get a similar behavior. Fig. 3 shows the spectra of the linear operators associated with KoVAE and *constrained* KoVAE. We can analyze and interpret the learned dynamics by investigating the spectrum. Specifically, if $|\lambda_j| < 1$, it is associated with exponentially decaying modes since $\lim_{t\to\infty}|\lambda^t_j| = 0$. When $|\lambda_j| = 1$, the associated mode has infinite memory, whereas $|\lambda_j| > 1$ is associated with unstable behavior of the dynamics since $\lim_{t\to\infty}|\lambda^t_j| = \infty$. In Fig. 3, we observe that training without the constraint results in decaying dynamics where all eigenvalues $\ll 1$ (yellow). In contrast, analyzing our approximate operator reveals that as expected, $|\lambda_p|$ and $|\lambda_q|$ (purple) are approximately one, indicating stable dynamics and the rest (yellow) are approximately zero, as desired. Finally, we also compute the spectra associated with the computed operators in the regular setting and discussed the results in App. F.

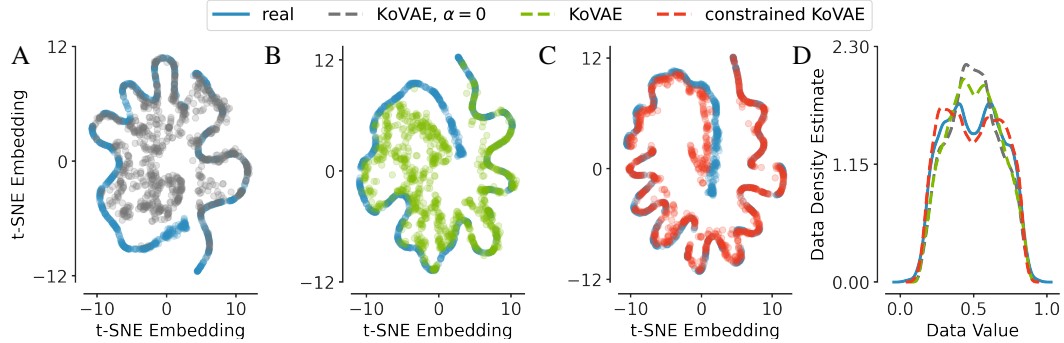

Figure 4: The `t-SNE` plots of KoVAE with $\alpha = 0$ (A), KoVAE (B), and constrained KoVAE (C), and their probability density functions (D) are compared to the true nonlinear pendulum data.

In addition to analyzing the spectrum, we visualize in Fig. 4 the `t-SNE` plots of the real data in comparison to the data generated using the three trained models, KoVAE with $\alpha = 0$ in Fig. 4A, KoVAE in Fig. 4B, and constrained KoVAE in Fig. 4C. Further, we also plot the probability density functions of the three models in Fig. 4D. The results clearly show that the synthetic samples generated with the constrained KoVAE (red) better match the distribution and PDF of the true process in comparison to KoVAE (green) and KoVAE with $\alpha = 0$ (gray).

## 5.3 ABLATION STUDY

We ablate our approach on the discriminative task in the regular and $30\%$ and $50\%$ irregular settings. We eliminate the linear prior by fixing $\alpha = 0$, and we also train another baseline without the linear prior and without the recurrent (GRU) component. Tab. 4 details our ablation results, and we observe that KoVAE outperforms all other regular ablation baselines and the majority of irregular baselines. We conclude that by introducing a novel linear dynamical prior, KoVAE improves generative results.

Table 4: Discriminative ablation results with regular and irregular $30\%$ and $50\%$ data on our method (KoVAE), without the linear prior ($\alpha = 0$), and also without the RNN ($\alpha = 0$, no GRU).

|  | regular | | | | 30% | | | | 50% | | | |
|---|---|---|---|---|---|---|---|---|---|---|---|---|
|  | Sines | Stocks | Energy | MuJoCo | Sines | Stocks | Energy | MuJoCo | Sines | Stocks | Energy | MuJoCo |
| KoVAE | **0.005** | **0.009** | **0.143** | **0.076** | **0.035** | **0.162** | **0.280** | **0.123** | **0.030** | **0.092** | **0.298** | 0.117 |
| $\alpha = 0$ | 0.006 | 0.023 | 0.155 | 0.087 | 0.038 | 0.172 | 0.295 | 0.131 | 0.040 | 0.181 | 0.306 | **0.108** |
| $\alpha = 0$, no GRU | - | - | - | - | 0.268 | 0.272 | 0.436 | 0.267 | 0.275 | 0.225 | 0.444 | 0.305 |

## 6 CONCLUSION

Generative modeling of time series data is often modeled with GANs which are unstable to train and exhibit mode collapse. In contrast, VAEs are more robust to such issues, however, current generative time series approaches employ non-sequential prior models. Further, introducing constraints that impose domain knowledge is challenging in these frameworks. We propose Koopman VAE (KoVAE), a new variational autoencoder that is based on a novel dynamical linear prior. Our method enjoys the benefits of VAEs, it supports regular and irregular time series data, and it facilitates the incorporation of physics-constraints, and analysis through the lens of linear dynamical systems theory. We extensively evaluate our approach on generative benchmarks in comparison to strong baselines, and we show that KoVAE significantly outperforms existing work on several quantitative and qualitative metrics. We also demonstrate on a real-world challenging climate dataset that KoVAE approximates well the associated density distribution and it generates accurate temperature long-term signals. Future work will explore the utility of KoVAE for scientific and engineering problems in more depth.

ACKNOWLEDGMENTS

OA was partially supported by an ISF grant 668/21, an ISF equipment grant, and by the Israeli Council for Higher Education (CHE) via the Data Science Research Center, Ben-Gurion University of the Negev, Israel. MWM would like to acknowledge NSF and ONR for providing partial support of this work. NBE would like to acknowledge NSF, under Grant No. 2319621, and the U.S. Department of Energy, under Contract Number DE-AC02-05CH11231, for providing partial support of this work. IN was partially supported by the Lynn and William Frankel Center of the Computer Science Department, Ben-Gurion University of the Negev. Our conclusions do not necessarily reflect the position or the policy of our sponsors, and no official endorsement should be inferred.

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

## A   STATIC VARIATIONAL AUTOENCODERS.

We recall a few governing equations from (Kingma & Welling, 2014; Doersch, 2016). In generative modeling, we want to compute the probability density function (PDF) $P(x)$ using a *maximum likelihood* framework. The density $P(x)$ reads

$$P(x) = \int p(x|z;\theta)p(z)\,\mathrm{d}z\,, \tag{12}$$

where $z$ is a latent representation associated with $x$. The model $p(x|z;\theta)$ ideally approaches $x$ for some $z$. In variational inference, we typically have that $p(x|z;\theta) = \mathcal{N}(x|f(z;\theta),\sigma^2 I)$, i.e., a Gaussian distribution where the mean is generated with a neural network $f$. However, two issues arise with the above PDF: 1) how to define $z$; and 2) how to integrate over $z$.

VAEs provide a solution for both issues. The first issue is resolved by assuming that $z$ is drawn from a predefined distribution, e.g., $\mathcal{N}(0, I)$. The second issue can be potentially solved by approximating Eq. 12 with a sum, i.e., $P(x) \approx \frac{1}{n}\sum_i p(x|z_i)$. A challenge, however, is that $n$ needs to be huge in high-dimensions. Instead, we observe that for most $z$, the term $p(x|z)$ will be nearly zero, and thus we can focus on sampling $z$ that are likely to produce $x$. Consequently, we need an approximate posterior $q(z|x)$ which allows to compute $\mathbb{E}_{z\sim q}p(x|z)$, where $z$ is obtained using the re-parametrization trick.

To guide training so that the prior and approximate posterior match, we employ the Kullback–Liebler Divergence $\mathrm{KL}[\cdot\,\|\,\cdot]$ between $q(z)$ and $p(z|x)$. The model is trained using the evidence lower bound (ELBO) loss $\mathbb{E}_{z\sim q}[\log p(x|z)] - \mathrm{KL}[q(z|x)\,\|\,p(z)]$, which is one of the core equations of VAE. Notice that the objective takes the form of reconstruction and regularization penalties, respectively.

## B   NEURAL CONTROLLED DIFFERENTIAL EQUATIONS

Irregularly sampled TS information $x_{t_1:t_N}$ for, e.g., $t_j \in [1,\ldots,T]$ cannot be modeled directly with discrete-time architectures such as recurrent neural networks (RNN). Therefore, the continuous analog of RNN is considered for $x_{t_1:t_N}$ based on neural controlled differential equations (NCDE) (Kidger et al., 2020) that are given by

$$h(t_{i+1}) = h(t_i) + \int_{t_i}^{t_{i+1}} f(h(t)\,;\,\theta_f)\,\mathrm{d}X(t)\,, \tag{13}$$

where $h(t_i)$ is the hidden code associated with $x_{t_i}$, $X(t)$ is a continuous-time trajectory generated from $x_{t_1:t_N}$ via an interpolation method with $X(t_i) := x_{t_i}$, and $f$ is a neural network parametrized by $\theta_f$ whose role is to learn the latent infinitesimal factor.

## C   LOSS FUNCTION EVALUATION

Our model is trained using the objective function $\mathcal{L}$, defined in Eq. 8, and repeated below,

$$\mathcal{L} = \mathbb{E}_{z_{1:T}\sim q}[\log p(x_{t_1:t_N}|z_{1:T})] + \alpha\mathcal{L}_{\mathrm{pred}}(z_{1:T},\bar{z}_{1:T}) - \beta\mathrm{KL}[q(z_{1:T}|x_{t_1:t_N})\,\|\,p(z_{1:T})]\,,$$

where the first addend is the reconstruction term, the second addend is the predictive loss term, and the last addend is the KL regularization term. To evaluate $\mathcal{L}$ in practice, we make a few standard assumptions, followed by straightforward computations. First, we assume that $p(x_{1:T}|z_{1:T})$ follows a Gaussian distribution whose mean is the output of the model and its variance is some constant. Namely, $p(x_{1:T}|z_{1:T}) = \mathcal{N}(x_{1:T}\,;\,\tilde{x}_{1:T}(z_{1:T},\theta),\sigma^2)$, where $\tilde{x}_{1:T}(z_{1:T})$ denotes the output of the decoder whose learnable parameters are given by $\theta$. This is a common probabilistic assumption (Goodfellow et al., 2016), under which the term $\mathbb{E}_{z_{1:T}\sim q}[\log p(x_{1:T}|z_{1:T})]$ becomes a simple mean squared error (MSE) between $x_{1:T}$ and $\tilde{x}_{1:T}$. Second, a similar reasoning is applied to $\mathcal{L}_{\mathrm{pred}}$, yielding an MSE evaluation between $z_{1:T}$ and $\bar{z}_{1:T}$. Finally, the regularization term involves the distribution $p(z_{1:T}) := \delta(z_{1:T} - A\bar{z}_{1:T})p(\bar{z}_{1:T})$, where $\delta(\cdot)$ is the Dirac delta distribution. However, the KL divergence is always evaluated in our framework on batches $z_{1:T}$ computed from $\bar{z}_{1:T}$. Therefore, we approximate $\delta(z_{1:T} - A\bar{z}_{1:T})$ by 1, leading to $p(z_{1:T}) \approx p(\bar{z}_{1:T})$. Then, we can compute the KL term using a closed-form formulation given the mean and variance of two distributions.

## D  VARIATIONAL PENALIZED EVIDENCE LOWER BOUND

In what follows, we present the derivation of the variational penalized evidence lower bound for our method. Our objective is to minimize $KL[q(z_{<T}|x_{\leq T})||p(z_{\leq T}|x_{\leq T})]$ under the Koopman constraint $\mathcal{L}_{\text{pred}}(z_{\leq T}, \bar{z}_{\leq T}) = \mathbb{E}_{\bar{z}_{\leq T} \sim p}[\log p(z_{\leq T}|\bar{z}_{\leq T})]$.

$$KL[q(z_{\leq T}|x_{\leq T})||p(z_{\leq T}|x_{\leq T})] + \mathbb{E}_{\bar{z}_{\leq T} \sim p}[\log p(z_{\leq T}|\bar{z}_{\leq T})]$$

$$= \int q(z_{\leq T}|x_{\leq T}) \log \frac{q(z_{\leq T}|x_{\leq T})}{p(z_{\leq T}|x_{\leq T})} dz_{\leq T} + \mathbb{E}_{\bar{z}_{\leq T} \sim p}[\log p(z_{\leq T}|\bar{z}_{\leq T})]$$

$$= \int q(z_{\leq T}|x_{\leq T}) \log \frac{q(z_{\leq T}|x_{\leq T})p(x_{\leq T})}{p(z_{\leq T}, x_{\leq T})} dz_{\leq T} + \mathbb{E}_{\bar{z}_{\leq T} \sim p}[\log p(z_{\leq T}|\bar{z}_{\leq T})]$$

$$= \int q(z_{\leq T}|x_{\leq T})[\log p(x_{\leq T}) + \log \frac{q(z_{\leq T}|x_{\leq T})}{p(z_{\leq T}, x_{\leq T})}] dz_{\leq T} + \mathbb{E}_{\bar{z}_{\leq T} \sim p}[\log p(z_{\leq T}|\bar{z}_{\leq T})]$$

$$= \log p(x_{\leq T}) + \int q(z_{\leq T}|x_{\leq T}) \log \frac{q(z_{\leq T}|x_{\leq T})}{p(z_{\leq T}, x_{\leq T})} dz_{\leq T} + \mathbb{E}_{\bar{z}_{\leq T} \sim p}[\log p(z_{\leq T}|\bar{z}_{\leq T})]$$

$$= \log p(x_{\leq T}) + \int q(z_{\leq T}|x_{\leq T}) \log \frac{q(z_{\leq T}|x_{\leq T})}{p(z_{\leq T})p(x_{\leq T}|z_{\leq T})} dz_{\leq T} + \mathbb{E}_{\bar{z}_{\leq T} \sim p}[\log p(z_{\leq T}|\bar{z}_{\leq T})]$$

$$= \log p(x_{\leq T}) + \int q(z_{\leq T}|x_{\leq T})[\log \frac{q(z_{\leq T}|x_{\leq T})}{p(z_{\leq T})} - \log p(x_{\leq T}|z_{\leq T})] dz_{\leq T} + \mathbb{E}_{\bar{z}_{\leq T} \sim p}[\log p(z_{\leq T}|\bar{z}_{\leq T})]$$

$$= \log p(x_{\leq T}) + \mathbb{E}_{z_{\leq T} \sim q(z_{\leq T}|x_{\leq T})}[\log \frac{q(z_{\leq T}|x_{\leq T})}{p(z_{\leq T})} - \log p(x_{\leq T}|z_{\leq T})] + \mathbb{E}_{\bar{z}_{\leq T} \sim p}[\log p(z_{\leq T}|\bar{z}_{\leq T})]$$

$$= \log p(x_{\leq T}) + KL[q(z_{\leq T}|x_{\leq T})||p(z_{\leq T})] - \mathbb{E}_{z_{\leq T} \sim q(z_{\leq T}|x_{\leq T})}[\log p(x_{\leq T}|z_{\leq T})] + \mathbb{E}_{\bar{z}_{\leq T} \sim p}[\log p(z_{\leq T}|\bar{z}_{\leq T})]$$

Thus we have the following:

$$KL[q(z_{\leq T}|x_{\leq T})||p(z_{\leq T}|x_{\leq T})] + \mathbb{E}_{\bar{z}_{\leq T} \sim p}[\log p(z_{\leq T}|\bar{z}_{\leq T})] =$$
$$\log p(x_{\leq T}) + KL[q(z_{\leq T}|x_{\leq T})||p(z_{\leq T})] - \mathbb{E}_{z_{\leq T} \sim q(z_{\leq T}|x_{\leq T})}[\log p(x_{\leq T}|z_{\leq T})] + \mathbb{E}_{\bar{z}_{\leq T} \sim p}[\log p(z_{\leq T}|\bar{z}_{\leq T})]$$

Now, rearranging the equation, we yield:

$$\log p(x_{\leq T}) - KL[q(z_{\leq T}|x_{\leq T})||p(z_{\leq T}|x_{\leq T})] + \mathbb{E}_{\bar{z}_{\leq T} \sim p}[\log p(z_{\leq T}|\bar{z}_{\leq T})] =$$
$$- \mathbb{E}_{z_{\leq T} \sim q(z_{\leq T}|x_{\leq T})}[\log p(x_{\leq T}|z_{\leq T})] + KL[q(z_{\leq T}|x_{\leq T})||p(z_{\leq T})] + \mathbb{E}_{\bar{z}_{\leq T} \sim p}[\log p(z_{\leq T}|\bar{z}_{\leq T})]$$

Thus finally:

$$\log p(x_{\leq T}) + \mathbb{E}_{\bar{z}_{\leq T} \sim p}[\log p(z_{\leq T}|\bar{z}_{\leq T})] \leq \tag{14}$$
$$- \mathbb{E}_{z_{\leq T} \sim q(z_{\leq T}|x_{\leq T})}[\log p(x_{\leq T}|z_{\leq T})] + KL[q(z_{\leq T}|x_{\leq T})||p(z_{\leq T})] + \mathbb{E}_{\bar{z}_{\leq T} \sim p}[\log p(z_{\leq T}|\bar{z}_{\leq T})]$$

# E    DATASETS AND BASELINE METHODS

We consider four synthetic and real-world datasets with different characteristic properties and statistical features, following challenging benchmarks on generative time series (Yoon et al., 2019; Jeon et al., 2022). **Sines**, is a multivariate simulated data, where each sample $x_t^i(j) := \sin(2\pi\eta t + \theta)$, with $\eta \sim \mathcal{U}[0,1]$, $\theta \sim \mathcal{U}[-\pi,\pi]$, and channel $j \in \{1,...,5\}$ for every $i$ in the dataset. This dataset is characterized by its continuity and periodic properties. **Stocks**, consists of the daily historical Google stocks data from 2004 to 2019 and has six channels including high, low, opening, closing, and adjusted closing prices and the volume. In contrast to Sines, Stocks is assumed to include random walk patterns and it is generally aperiodic. **Energy**, is a multivariate UCI appliance energy prediction dataset (Candanedo, 2017), with 28 channels, correlated features, and it is characterized by noisy periodicity and continuous-valued measurements. Finally, **MuJoCo** (**Mu**lti-**Jo**int dynamics with **Co**ntact) (Todorov et al., 2012), is a general-purpose physics generator, which we use to simulate time series data with 14 channels.

We compare our method with several state-of-the-art (SOTA) generative time series models, such as TimeGAN (Yoon et al., 2019), RCGAN (Esteban et al., 2017), C-RNN-GAN (Mogren, 2016), WaveGAN (Donahue et al., 2019), WaveNet (van den Oord et al., 2016), T-Forcing (Graves, 2013), P-Forcing (Goyal et al., 2016), TimeVAE (Desai et al., 2021), CR-VAE (Li et al., 2023), and the recent irregular method GT-GAN (Jeon et al., 2022). All the methods, except for GT-GAN, are not designed to handle missing observations; thus, we follow GT-GAN and compare them with their re-designed versions. Specifically, extending regular approaches to support irregular TS requires the conversion of a dynamical module to its time-continuous version. For instance, we converted GRU layers to GRU-$\Delta t$ and GRU-D to exploit the time difference between observations and to learn the exponential decay between samples, respectively. We denote the re-designed methods by adding $\Delta t$ or D postfix to their name, such as TimeGAN-$\Delta t$ and TimeGAN-D.

# F    CONSTRAINED GENERATION AND ANALYSIS

In addition to the analysis of the constrained system in Eq. 10, we also compute the approximate Koopman operators for all the regular systems we consider in the main text. Specifically, we show in Fig. 5 four panels, corresponding to the spectral eigenvalue plots for Sines (left), Stocks (middle left), Energy (middle right), and MuJoCo (right). Green eigenvalues are within the unit circle, whereas red eigenvalues are located outside the unit circle. Noticeably, the learned dynamics for Sines, Stocks and MuJoCo are stable, i.e., their corresponding eigenvalues are within the unit circle, and thus, their training, inference and overall long-term behavior is expected to be numerically stable and not produce values that present large deviations. In contrast, the spectrum associated with the model learned for the Energy dataset reveals unstable dynamics where the largest $\lambda_i \gg 1$. We hypothesize that we can benefit from incorporating spectral constraints to regularize the dynamics and achieve stable dynamics. In the future, we will investigate whether spectral constraints can enhance the generation results and overall behavior of such models.

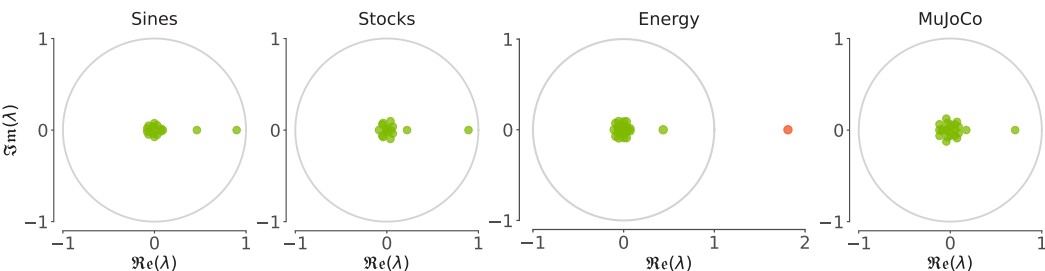

Figure 5: The spectral distribution of the approximate Koopman operator of the prior for each dataset in the regular setting.

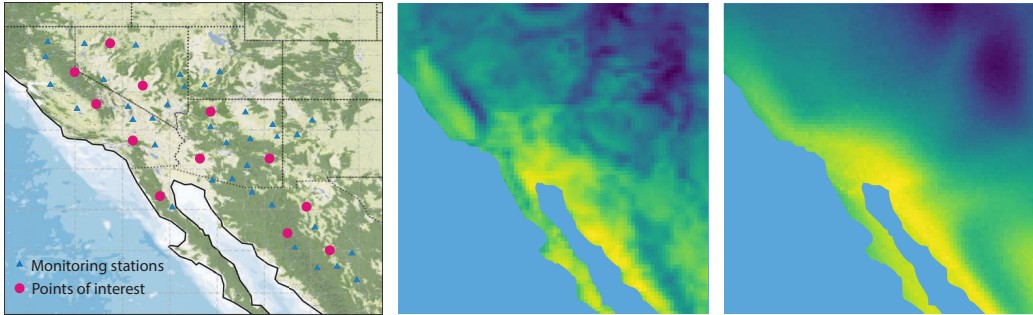

Figure 6: Comparison of the temporally-averaged temperature distribution of the generated signals and ground truth data in California region. left: California region map. middle: ground truth average. right: generated average.

## G    CONDITIONAL GENERATION FOR WEATHER DATA

Accurate weather data is essential for various applications, such as agriculture, water resource management, and disaster early warning. However, due to the limited availability of weather stations, scientists are faced with the challenge of the inherent sparsity of observational weather data, especially in specific geographical areas of interest. Generative models can produce synthetic weather data to supplement sparse or incomplete observational datasets. In this paper, we explore the potential of our KoVAE model for modeling temperature dynamics based on sparse measurement data. Our objective is to generate the temperature data at any location within a specific region of interest. Hence, the proposed KoVAE model here is conditioned on the geospatial coordinates of the specific region. By leveraging conditional generative models, we hope to provide valuable insights into weather dynamics in specific regions and enhance the decision-making processes that are reliant on accurate weather information. Our model is trained with geospatial coordinates as conditional variables. To support the conditional training and generation, we made two simple modifications: (i) we add a very simple MLP to support spatial embeddings $s$; and (ii) we augment the decoder and prior to support conditional generation: $p(x_{t_1:t_N}|z_{1:T}, s)$ and $p(x_{1:T}, z_{1:T}, s) = p(z_{1:T}|s)p(x_{1:T}|z_{1:T}, s)$.

We focus on two representative regions in the United States: California and Central America areas. California has diverse and non-stationary weather changes due to the complex interactions between seas, lands, and mountains. Central America areas present more stable and relatively easier temperature dynamics compared with the California region. Both of these regions include a grid of $80 \times 80$, which means we have 6400 time series samples in total. Moreover, the dataset in this task comprises temperature at 2-meter height above the surface from ERA5 reanalysis dataset (Hersbach et al., 2020). We select 4-month observation data within the specific area for training and evaluating our VAE model. It contains 120 time steps for each time series sample. We split the train and test sets with a ratio of $80\%$ and $20\%$. The generated time series samples are shown in Fig. 9 and Fig. 10. We can see that our KoVAE model can accurately capture the underlying dynamics of temperature data in both California and Central America regions. Furthermore, Fig. 6 and Fig. 7 exhibit the comparisons of the temporally-averaged temperature distribution between the generations and ground truth in California and Central America regions. The left panels in these figures show the points of interest with respect to the given monitoring stations, which can be used to condition our model. The temperature distribution patterns of the generations align well with those of the ground truth in the entire domain. Similarly, we also observe that the generation and ground truth match well when comparing minimum and maximum values over time, as shown in Fig. 8. Nevertheless, the fine-scale spatial details are lacking in our generation due to the absence of spatial constraints. In the future, we will enhance our conditional KoVAE model by incorporating spatial continuity and prior knowledge for spatiotemporal generation.

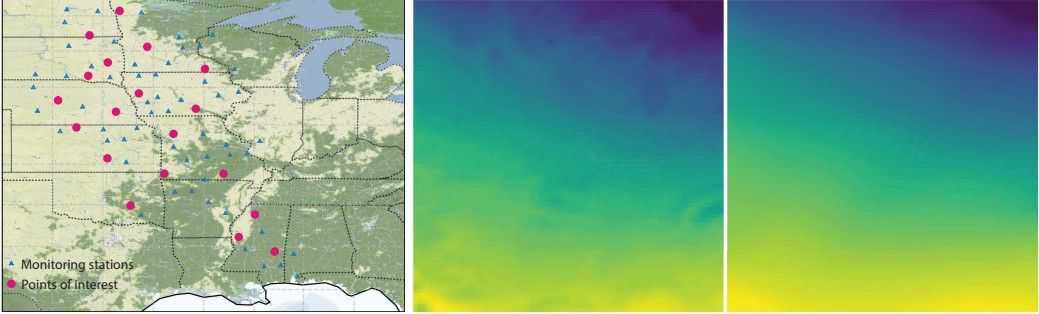

Figure 7: Comparison of the temporally-averaged temperature distribution between the generations and ground truth in Central America region. left: Central America region map. middle: ground truth average. right: generated average.

## H  ADDITIONAL GENERATION RESULTS

For brevity, in the main text, we provided tables without standard deviation (std). In this section, we provide extended tables that include the std for each metric. Tab. 6 and Tab. 7 present the results for the regular observed data of discriminative and predictive scores, respectively. In Tab. 8, Tab. 9, and Tab. 10 we show the results for the irregular sampled of 30%, 50%, and 70% missing observation, respectively. Each table includes both discriminative and predictive scores. Furthermore, we provide qualitative results to compare our model with GT-GAN, the second-best generation model for both regularly and irregularly sampled data. In Fig. 11 Fig. 12, Fig. 13, we visualize the t-SNE projection of both ground truth data and generated data (first row) and the PDF of ground truth data vs. generated data (second row). We perform the visualization for all regularly and irregularly sampled datasets.

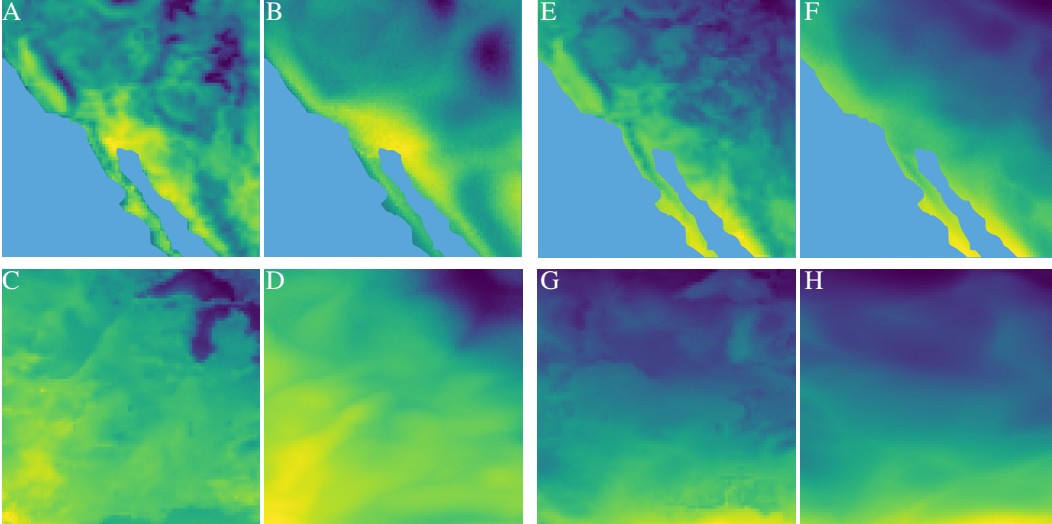

Figure 8: The maximum and minimum plots for both California and Central America regions. (A) The maximum value over time of the ground truth in the California region. (B) The maximum value over time of the generation in the California region. (C) The maximum value over time of the ground truth in the Central America region. (D) The maximum value over time of the generation in the Central America region. (E) The minimum value over time of the ground truth in the California region. (F) The minimum value over time of the generation in the California region. (G) The minimum value over time of the ground truth in the Central America region. (H) The minimum value over time of the generation in the Central America region.

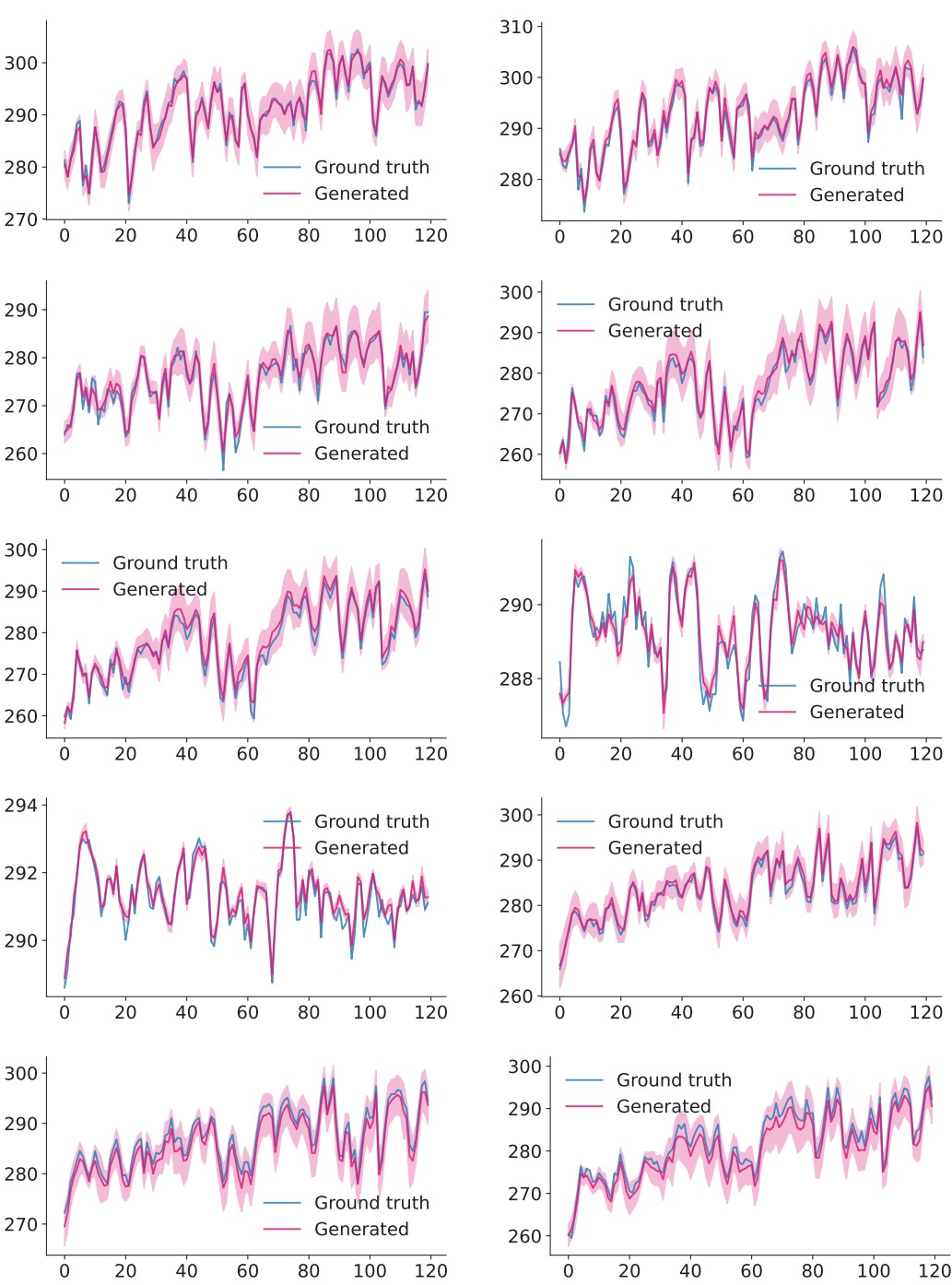

Figure 9: The representative generations in the California area.

# I    COMPUTATIONAL RESOURCES

Here we compare the model complexity (number of parameters), and the wall clock time measurement with GT-GAN. We present the details in Tab. 5. Both models were trained on the same software and hardware for fair comparison. The software environments we use are: CentOS Linux 7 (Core) and PYTHON 3.9.16, and the hardware is: NVIDIA RTX 3090. We observe that we use

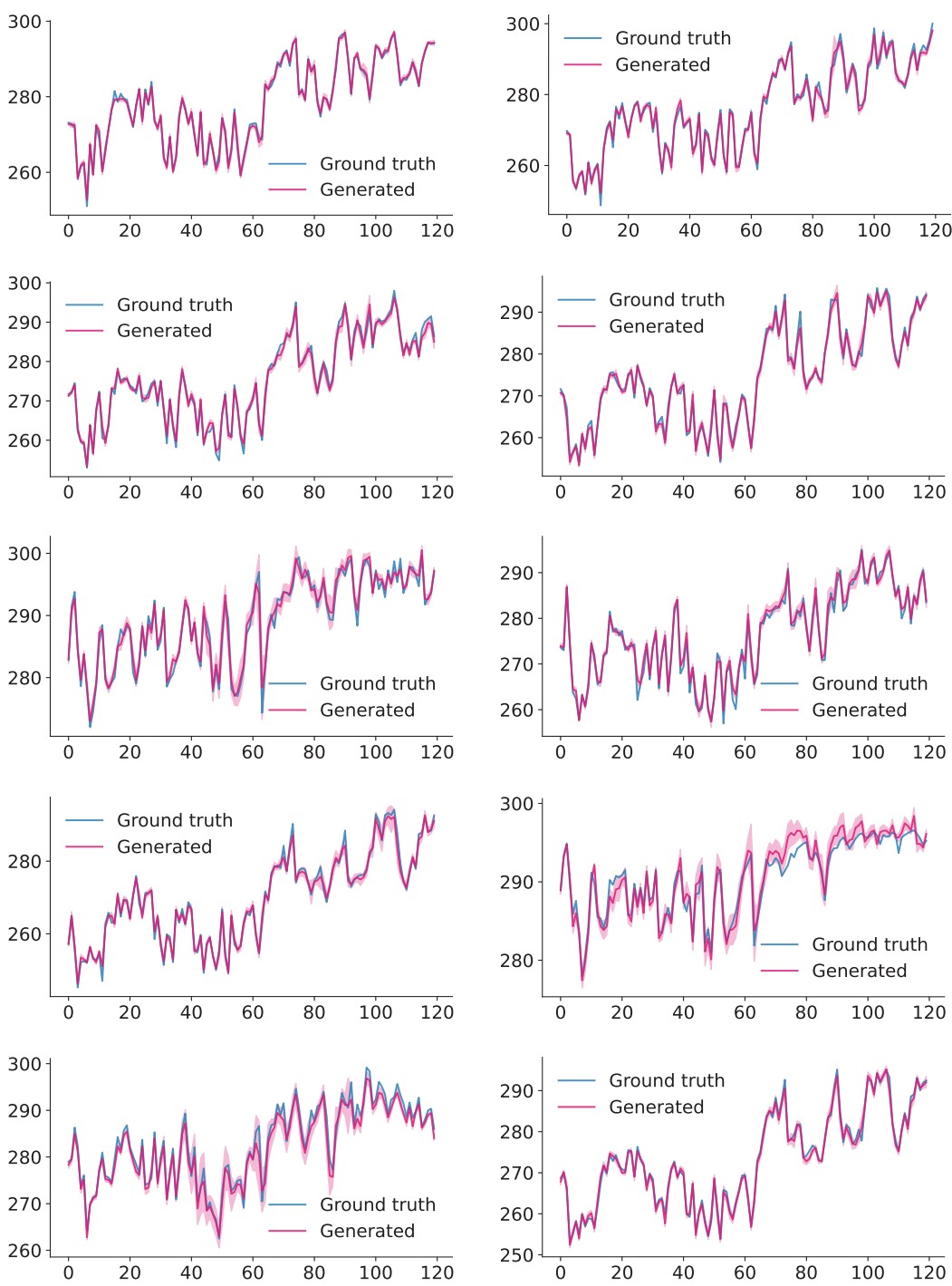

Figure 10: The representative generations in the Central America area.

more parameters on the Energy and MuJoCo datasets, but the training time is significantly faster on the same software and hardware.

| Method | Copm. | | Sines | Stocks | Energy | MuJoCo |
|---|---|---|---|---|---|---|
| KoVAE (Ours) | # Parameters | | **32,929** | **33,410** | 230,860 | 195,902 |
| | Wall Clock Time | | **2h 49m** | **2h 49m** | **5h 16m** | **3h 46m** |
| GT-GAN | # Parameters | | 41,913 | 41,776 | **57,104** | **47,346** |
| | Wall Clock Time | | 10h 12m | 12h 20m | 10h 39m | 13h 12m |

Table 5: Comparison of number (#) of parameters and wall clock time.

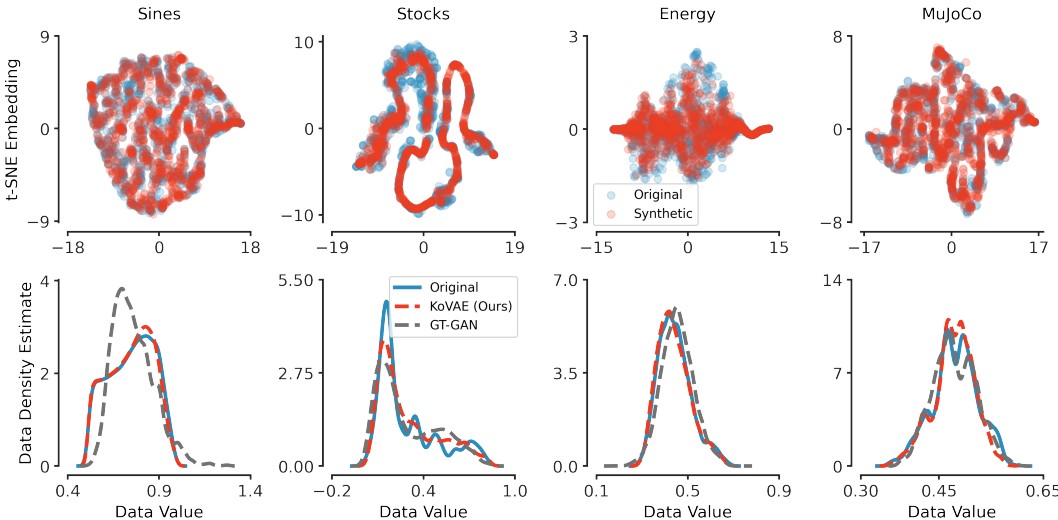

Figure 11: We qualitatively evaluate our approach on regularly sampled data with two-dimensional t-SNE plots of the synthetic and real data (top row). In addition, we show the probability density functions of the real data, and for KoVAE and GT-GAN synthetic distributions (bottom row).

## J    HYPERPARAMETERS ROBUSTNESS

We also explore how stable our model is to hyperparameter choice. To this end, we perform an extensive grid search over the following space for:

$$\alpha, \beta \in \{1.0, 0.9, 0.7, 0.5, 0.3, 0.1, 0.09, 0.07, 0.05, 0.03, 0.01,$$
$$0.009, 0.007, 0.005, 0.003, 0.001, 0.0009, 0.0007, 0.0005\}^2$$

for the stocks dataset. Fig. 14 shows the discriminative score for each combination of $\alpha$ and $\beta$. Most of the values are lower than the second-best model for this task.

## K    RECONSTRUCTION RESULTS

In addition to studying the generative capabilities of KoVAE, we also investigate its reconstruction and inference features below. We show in Fig. 15 the reconstructed signals in the regular setting. Each subplot represent a separate feature, where for datasets with more than five channels, we plot the first five features. Solid lines represent ground-truth data, whereas dashed lines are the reconstructions our model outputs. For simplicity, we omitted axis labels. For all subplots, the $x$-axis represents time, and the $y$-axis is the feature values. Moreover, we also plot in Fig. 16 the inferred signals in the irregular 50% setting. In this case, half of the signal is removed during training, and thus, the reconstructions present the inference capabilities of our model. Our results indicate that KoVAE is able to successfully reconstruct and infer the data.

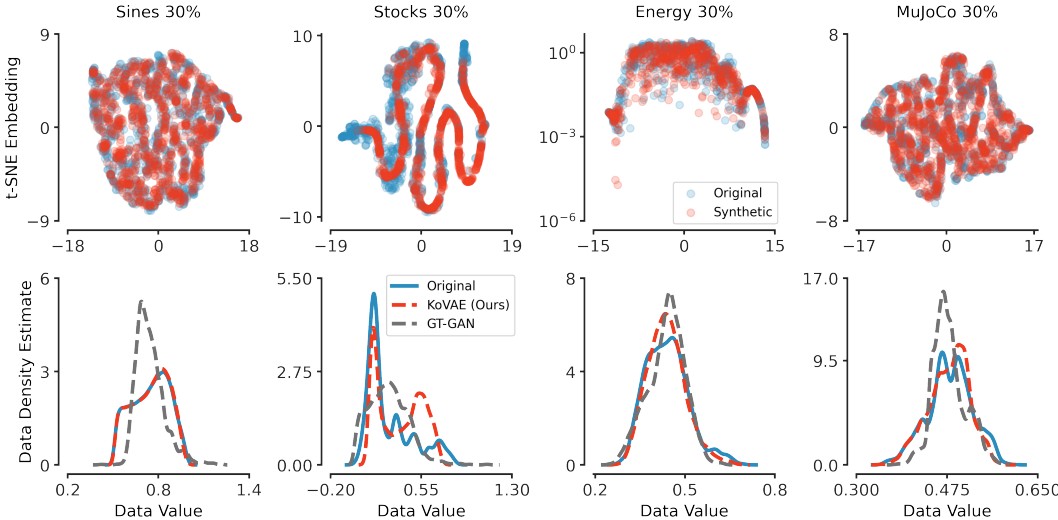

Figure 12: We qualitatively evaluate our approach on 30% irregularly sampled data with two-dimensional t-SNE plots of the synthetic and real data (top row). In addition, we show the probability density functions of the real data, and for KoVAE and GT-GAN synthetic distributions (bottom row).

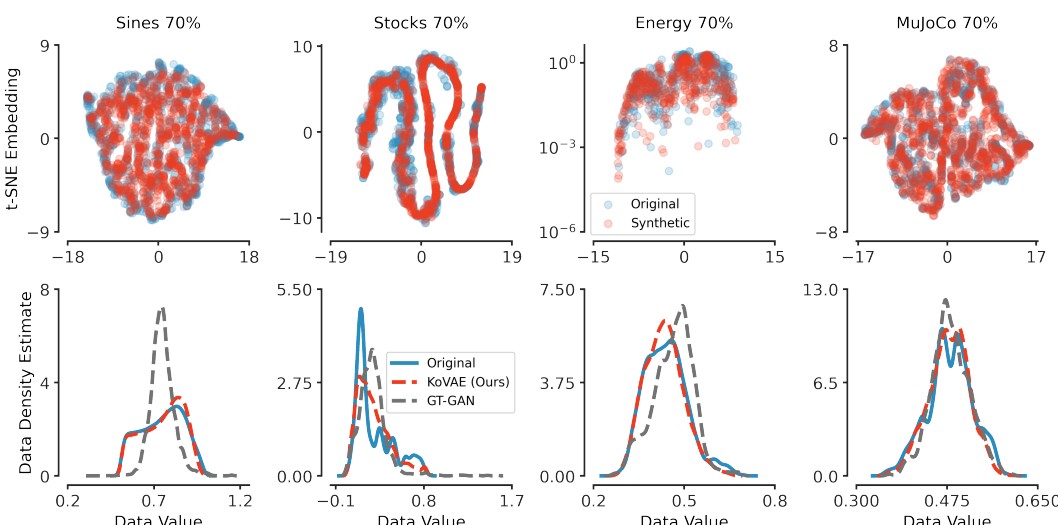

Figure 13: We qualitatively evaluate our approach on 70% irregularly sampled data with two-dimensional t-SNE plots of the synthetic and real data (top row). In addition, we show the probability density functions of the real data, and for KoVAE and GT-GAN synthetic distributions (bottom row).

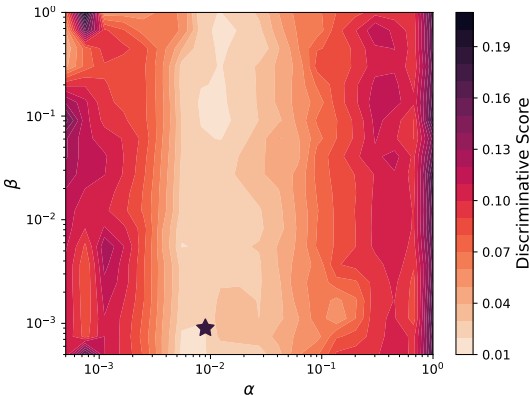

Figure 14: Each point in this plot represents the discriminative score for stocks regularly sampled data set when trained with specific $\alpha$ and $\beta$ values. The star marker denotes the best discriminative score for some $\alpha = 9\mathrm{e}{-3}, \beta = 9\mathrm{e}{-4}$ values. In addition, most of the values in the figure present better performance than the second-best method.

Table 6: Regular TS, discriminative task

| Method | Sines | Stocks | Energy | MuJoCo |
|---|---|---|---|---|
| KoVAE (Ours) | **.005±.003** | **.009±.006** | **.143±.011** | **.076±.017** |
| GT-GAN | .012±.014 | .077±.031 | .221±.068 | .245±.029 |
| TimeGAN | .011±.008 | .102±.021 | .236±.012 | .409±.028 |
| TimeVAE | .016±.010 | .036±.033 | .323±.029 | .224±.026 |
| CR-VAE | .342±.157 | .320±.095 | .475±.054 | .464±.012 |
| RCGAN | .022±.008 | .196±.027 | .336±.017 | .436±.012 |
| C-RNN-GAN | .229±.040 | .399±.028 | .499±.001 | .412±.095 |
| T-Forcing | .495±.001 | .226±.035 | .483±.004 | .499±.000 |
| P-Forcing | .430±.227 | .257±.026 | .412±.006 | .500±.000 |
| WaveNet | .158±.011 | .232±.028 | .397±.010 | .385±.025 |
| WaveGAN | .277±.013 | .217±.022 | .363±.012 | .357±.017 |
| RI | **54.54%** | **75.00%** | **35.29%** | **66.07%** |

Table 7: Regular TS, predictive task

| Method | Sines | Stocks | Energy | MuJoCo |
|---|---|---|---|---|
| KoVAE (Ours) | **.093±.000** | **.037±.000** | **.251±.000** | **.038±.002** |
| GT-GAN | .097±.000 | .040±.000 | .312±.002 | .055±.000 |
| TimeVAE | **.093±.000** | .037±.033 | .254±.000 | .039±.002 |
| TimeGAN | .093±.019 | .038±.001 | .273±.004 | .082±.006 |
| CR-VAE | .143±.002 | .076±.013 | .277±.001 | .050±.000 |
| RCGAN | .097±.001 | .040±.001 | .292±.005 | .081±.003 |
| C-RNN-GAN | .127±.004 | .038±.000 | .483±.005 | .055±.004 |
| T-Forcing | .150±.022 | .038±.001 | .315±.005 | .142±.014 |
| P-Forcing | .116±.004 | .043±.001 | .303±.006 | .102±.013 |
| WaveNet | .117±.008 | .042±.001 | .311±.005 | .333±.004 |
| WaveGAN | .134±.013 | .041±.001 | .307±.007 | .324±.006 |
| Original | 0.094 | 0.036 | 0.250 | 0.031 |

Table 8: Irregular time series (30% dropped)

| | Method | Sines | Stocks | Energy | MuJoCo |
|---|---|---|---|---|---|
| Discriminative Score | VKAE (Ours) | **.035**±**.023** | **.162**±**.068** | **.280**±**.018** | **.123**±**.018** |
| | GT-GAN | .363±.063 | .251±.097 | .333±.063 | .249±.035 |
| | TimeGAN-$\triangle t$ | .494±.012 | .463±.020 | .448±.027 | .471±.016 |
| | RCGAN-$\triangle t$ | .499±.000 | .436±.064 | .500±.000 | .500±.000 |
| | C-RNN-GAN -$\triangle t$ | .500±.000 | .500±.001 | .500±.000 | .500±.000 |
| | T-Forcing-$\triangle t$ | .395±.063 | .305±.002 | .477±.011 | .348±.041 |
| | P-Forcing-$\triangle t$ | .344±.127 | .341±.035 | .500±.000 | .493±.010 |
| | TimeGAN-D | .496±.008 | .411±.040 | .479±.010 | .463±.025 |
| | RCGAN-D | .500±.000 | .500±.000 | .500±.000 | .500±.000 |
| | C-RNN-GAN-D | .500±.000 | .500±.000 | .500±.000 | .500±.000 |
| | T-Forcing-D | .408±.087 | .409±.051 | .347±.046 | .494±.004 |
| | P-Forcing-D | .500±.000 | .480±.060 | .491±.020 | .500±.000 |
| Predictive Score | KoVAE (Ours) | **.074**±**.005** | **.019**±**.001** | **.049**±**.002** | **.043**±**.000** |
| | GT-GAN | .099±.004 | .021±.003 | .066±.001 | .048±.001 |
| | TimeGAN-$\triangle t$ | .145±.025 | .087±.001 | .375±.011 | .118±.032 |
| | RCGAN-$\triangle t$ | .144±.028 | .181±.014 | .351±.056 | .433±.021 |
| | C-RNN-GAN-$\triangle t$ | .754±.000 | .091±.007 | .500±.000 | .447±.000 |
| | T-Forcing-$\triangle t$ | .116±.002 | .070±.013 | .251±.000 | .056±.001 |
| | P-Forcing-$\triangle t$ | .102±.002 | .083±.018 | .255±.001 | .089±.011 |
| | TimeGAN-D | .192±.082 | .105±.053 | .248±.024 | .098±.006 |
| | RCGAN-D | .388±.113 | .523±.020 | .409±.020 | .361±.073 |
| | C-RNN-GAN-D | .664±.001 | .345±.002 | .440±.000 | .457±.001 |
| | T-Forcing-D | .100±.002 | .027±.002 | .090±.001 | .100±.001 |
| | P-Forcing-D | .154±.004 | .079±.008 | .147±.001 | .173±.002 |
| | Original** | .071±.004 | .011±.002 | .045±.001 | .041±.002 |

Table 9: Irregular time series (50% dropped)

| | Method | Sines | Stocks | Energy | MuJoCo |
|---|---|---|---|---|---|
| Discriminative Score | KoVAE (Ours) | **.030**±**.017** | **.092**±**.075** | **.298**±**.013** | **.117**±**.019** |
| | GT-GAN | .372±.128 | .265±.073 | .317±.010 | .270±.016 |
| | TimeGAN-$\triangle t$ | .496±.008 | .487±.019 | .479±.020 | .483±.023 |
| | RCGAN-$\triangle t$ | .406±.165 | .478±.049 | .500±.000 | .500±.000 |
| | C-RNN-GAN-$\triangle t$ | .500±.000 | .500±.000 | .500±.000 | .500±.000 |
| | T-Forcing -$\triangle t$ | .408±.137 | .308±.010 | .478±.011 | .486±.005 |
| | P-Forcing-$\triangle t$ | .428±.044 | .388±.026 | .498±.005 | .491±.012 |
| | TimeGAN-D | .500±.000 | .477±.021 | .473±.015 | .500±.000 |
| | RCGAN-D | .500±.000 | .500±.000 | .500±.000 | .500±.000 |
| | C-RNN-GAN-D | .500±.000 | .500±.000 | .500±.000 | .500±.000 |
| | T-Forcing-D | .430±.101 | .407±.034 | .376±.046 | .498±.001 |
| | P-Forcing-D | .499±.000 | .500±.000 | .500±.000 | .500±.000 |
| Predictive Score | KoVAE (Ours) | **.072**±**.002** | .019±.001 | **.049**±**.001** | **.042**±**.001** |
| | GT-GAN | .101±.010 | **.018**±**.002** | .064±.001 | .056±.003 |
| | TimeGAN-$\triangle t$ | .123±.040 | .058±.003 | .501±.008 | .402 ±.021 |
| | RCGAN-$\triangle t$ | .142±.005 | .094±.013 | .391±.014 | .277±.061 |
| | C-RNN-GAN-$\triangle t$ | .741±.026 | .089±.001 | .500±.000 | .448±.001 |
| | T-Forcing-$\triangle t$ | .379±.029 | .075±.032 | .251±.000 | .069±.002 |
| | P-Forcing-$\triangle t$ | .120±.005 | .067±.014 | .263±.003 | .189±.026 |
| | TimeGAN-D | .169±.074 | .254±.047 | .339±.029 | .375±.011 |
| | RCGAN-D | .519±.046 | .333±.044 | .250±.010 | .314±.023 |
| | C-RNN-GAN-D | .754±.000 | .273±.000 | .438±.000 | .479±.000 |
| | T-Forcing-D | .104±.001 | .038±.003 | .090±.000 | .113±.001 |
| | P-Forcing-D | .190±.002 | .089±.010 | .198±.005 | .207±.008 |
| | Original | .071±.004 | .011±.002 | .045±.001 | .041±.002 |

Table 10: Irregular time series (70% dropped)

|  | Method | Sines | Stocks | Energy | MuJoCo |
|---|---|---|---|---|---|
| **Discriminative Score** | KoVAE (Ours) | **.065**±**.012** | **.101**±**.040** | **.392**±**.004** | **.119**±**.014** |
|  | GT-GAN | .278±.022 | .230±.053 | **.325**±**.047** | .275±.023 |
|  | TimeGAN-△t | .500±.000 | .488±.009 | .496±.008 | .494±.009 |
|  | RCGAN-△t | .433±.142 | .381±.086 | .500±.000 | .500±.000 |
|  | C-RNN-GAN-△t | .500±.000 | .500±.000 | .500±.000 | .500±.000 |
|  | T-Forcing-△t | .374±.087 | .365±.027 | .468±.008 | .428±.022 |
|  | P-Forcing-△t | .288±.047 | .317±.019 | .500±.000 | .498±.003 |
|  | TimeGAN-D | .498±.006 | .485±.022 | .500±.000 | .492±.009 |
|  | RCGAN-D | .500±.000 | .500±.000 | .500±.000 | .500±.000 |
|  | C-RNN-GAN-D | .500±.000 | .500±.000 | .500±.000 | .500±.000 |
|  | T-Forcing-D | .436±.067 | .404±.068 | .336±.032 | .493±.005 |
|  | P-Forcing-D | .500±.000 | .449±.150 | .494±.011 | .499±.000 |
| **Predictive Score** | KoVAE (Ours) | **.076**±**.004** | **.012**±**.000** | **.052**±**.001** | **.044**±**.001** |
|  | GT-GAN | .088±.005 | .020±.005 | .076±.001 | .051±.001 |
|  | TimeGAN-△t | .734±.000 | .072±.000 | .496±.000 | .442±.000 |
|  | RCGAN-△t | .218±.072 | .155±.009 | .498±.000 | .222±.041 |
|  | C-RNN-GAN-△t | .751±.014 | .084±.002 | .500±.000 | .448±.001 |
|  | T-Forcing-△t | .113±.001 | .070±.022 | .251±.000 | .053±.002 |
|  | P-Forcing-△t | .123±.004 | .050±.002 | .285±.006 | .117±.034 |
|  | TimeGAN-D | .752±.001 | .228±.000 | .443±.000 | .372±.089 |
|  | RCGAN-D | .404±.034 | .441±.045 | .349±.027 | .420±.056 |
|  | C-RNN-GAN-D | .632±.001 | .281±.019 | .436±.000 | .479±.001 |
|  | T-Forcing-D | .102±.001 | .031±.002 | .091±.000 | .114±.003 |
|  | P-Forcing-D | .278±.045 | .107±.009 | .193±.006 | .191±.005 |
|  | Original | .071±.004 | .011±.002 | .045±.001 | .041±.002 |

Figure 15: We plot the original signals (solid lines) vs. reconstructed signals (dashed lines) for Sine, Stock, Energy, and Mujoco in the regular setting. Overall, our model matches the data well.

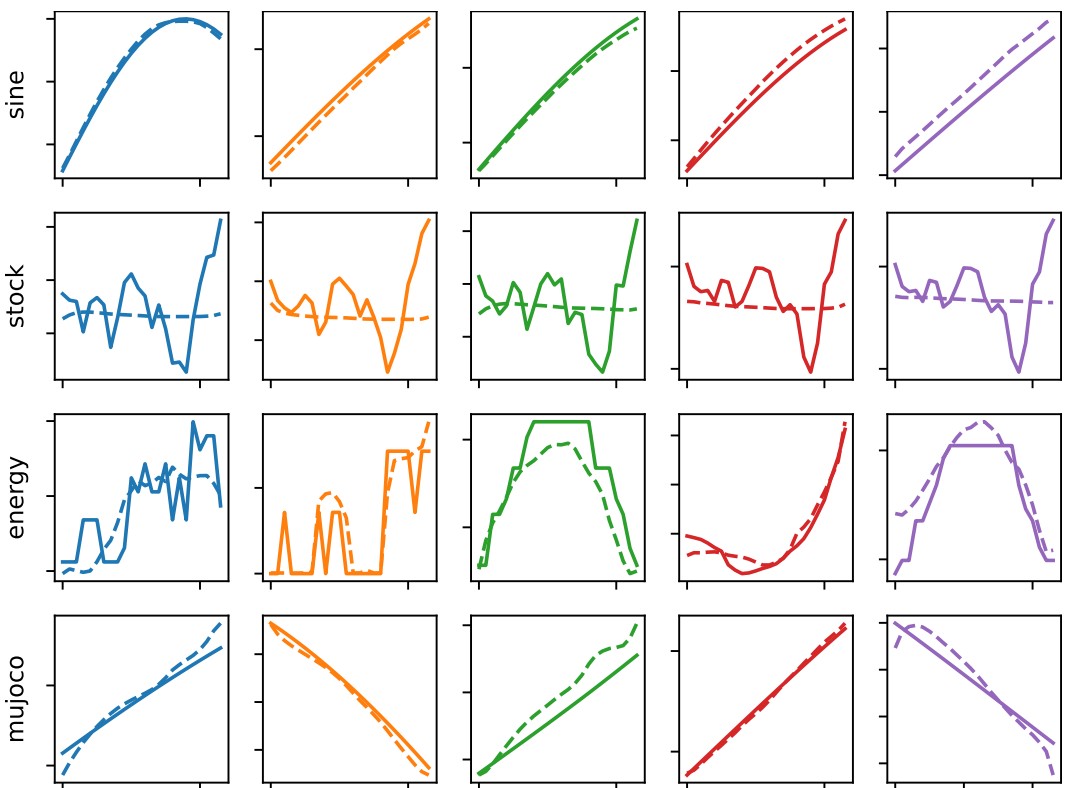

Figure 16: We plot the original signals (solid lines) vs. inferred signals (dashed lines) for Sine, Stock, Energy, and Mujoco in the irregular 50% case. Overall, our model recovers the data well, except for stock, where the output signal represents the average.

