# OpenReview forum: "Generative Modeling of Regular and Irregular Time Series Data via Koopman VAEs"
_ICLR.cc/2024/Conference — ICLR 2024 poster_

### Official Review · Reviewer_nGir · 2023-10-30

**Soundness:** 4 excellent
**Presentation:** 3 good
**Contribution:** 4 excellent
**Rating:** 8
**Confidence:** 5

**Summary:**

Although the existing GAN-based time-series generation showed good performance, they proposed a Koopman-based VAE model, citing problems such as unstable training or mode collapse. In addition, KVAE is a relatively simple model but it shows better performance in regular and irregular time-series than baseline models.

**Strengths:**

Originality: There was an existing model that applied the VAE model to time-series generation, but its performance was lacking. This paper showed good performance in time-series generation by applying Koopman to VAE. In addition, while the existing GAN-based model had many hyperparameters due to unstable training, KVAE is very simple and shows good performance.

Quality: In addition to various experiments conducted in previous research, the effectiveness of KAVE is clearly demonstrated through additional experiments such as physics-constrained generation.

Clarity: Very well written and easy to read.

Significance: Time-series generation suffered from many problems due to complex training. However, in this paper, it shows good performance with very easy learning.

**Weaknesses:**

1. There is a lack of explanation about Koopman in the Background section. In the case of this paper, the main point is Koopman, and readers may also want to know more about Koopman. Therefore, an explanation of Koopman should be in the main paper.

2. There are no results for the predictive loss term in the ablation study in Section 5.5.

Minor issues
1. For Section 5, Experiments seems to be a more appropriate word than Results.
2. It seems that 0 should be excluded from the MuJoCo results in Table 10.

**Questions:**

I am curious about the role of the predictive loss term. In this paper, a predictive loss term was added to the object function. Therefore, I am curious about how much the predictive loss term affects the performance that is superior to existing baseline models.

---

> ### Author Response · Authors · 2023-11-17
>
> We express our gratitude to Reviewer nGir for their favorable remarks regarding the simplicity of our model, the superior effectiveness of our approach, and the clarity of our writing. Furthermore, we appreciate their valuable contributions in posing significant questions and offering constructive suggestions for enhancing the quality of our paper. In what follows, we address the specific comments put forth by Reviewer nGir. Given the chance, we will be happy to integrate the suggested modifications outlined below into the final revision.
>
> >There is a lack of explanation about Koopman in the Background section. In the case of this paper, the main point is Koopman, and readers may also want to know more about Koopman. Therefore, an explanation of Koopman should be in the main paper.
>
> Following the reviewer's suggestion, we moved the appendix section that gives background information on Koopman theory and practice into Section 3 (Background) in the main paper.
>
> >There are no results for the predictive loss term in the ablation study in Section 5.5.
>
> Regrettably, there seems to be a typo in our ablation results in Table 4. Specifically, the submitted paper mentions $\beta$, suggesting that we removed the KL divergence term that minimizes the (pseudo-)distance between the prior and posterior distributions. However, in our ablation study, we tested the effect of the predictive loss term and the obtained results without utilizing this component. Thus, all appearances of $\beta$ in the ablation study should be replaced with $\alpha$.
>
> >For Section 5, Experiments seems to be a more appropriate word than Results.
>
> Following the reviewer's suggestion, we changed the section heading of Section 5 to read Experiments.
>
> >It seems that 0 should be excluded from the MuJoCo results in Table 10.
>
> Thank you for noticing this. We fixed it.
>
> >I am curious about the role of the predictive loss term. In this paper, a predictive loss term was added to the object function. Therefore, I am curious about how much the predictive loss term affects the performance that is superior to existing baseline models.
>
> Indeed, ablating the effect of the predictive loss term is important. This ablation appears in Table 4 (after correcting the typo $\beta \rightarrow \alpha$). The results indicate that the benefit of the predictive loss term can be significant. Notable examples are stocks regular, energy $30\%$, and stocks $50\%$. More generally, removing this loss term results in inferior error estimates in most cases.

---

> > ### Comment · Reviewer_nGir · 2023-11-22
> >
> > Thank you for the response to my questions.
> > I also looked at other reviewers' questions and the author's responses to them.
> > I have no further questions and will raise my score.

---

> > > ### Author Response · Authors · 2023-11-22
> > >
> > > Thank you for your feedback and for your willingness to raise your score.

---

> > > > ### Author Response · Authors · 2023-11-23
> > > >
> > > > We noticed your kind intention to revise the score, and we would greatly appreciate if you could take a moment to update the score as per your recent considerations.

---

> > > > > ### Comment · Reviewer_nGir · 2023-11-23
> > > > >
> > > > > The discussions above are excellent, and I wish I could see them in the revised manuscript, which I think would help readers better understand the contributions of this article.
> > > > >
> > > > > Given that my major concerns have already been addressed, I am now happy to update my rating to 8. Best of luck to the authors!

---

### Official Review · Reviewer_HuLD · 2023-10-30

**Soundness:** 2 fair
**Presentation:** 1 poor
**Contribution:** 2 fair
**Rating:** 5
**Confidence:** 5

**Summary:**

In this study, the authors propose a generative model using a variational auto-encoder. The variational auto-encoder employs the neural controlled differential equations to consider irregular time series and uses a GRU to march in time. The authors proposed to use a prior distribution with a linear transition function. It is shown that the proposed model outperforms previous models in the generation task.

**Strengths:**

The authors proposed to use a linear operator for the prior in the training of the variational auto-encoder. The proposed method is relatively straightforward and it is shown that the method outperforms some of the existing models for generation tasks.

**Weaknesses:**

The novelty of the study is limited. While the authors claim their method is based on the Koopman operator, their model is, in fact, more similar to the dynamic linear model, or Kalman filter. Also, one of the claims is that the model can take care of irregular time series, the capability is simply using a pre-existing model, neural controlled differential equations. Moreover, the manuscript is not well written. The probabilistic description and the models are not clearly defined. There are a few concerns about their model. See the comments below,

**Questions:**

1. I know trying to optimizer both the prior and posterior in a loss function has become a trend in the deep learning community. However, theoretically speaking, trying to optimize the prior and posterior jointly in the KL loss leads to a ill-posed problem, where a unique solution does not exist. Simply put, it becomes a ping-pong game between the prior and posterior. You can easily show it by computing the parameters of the distributions in a local minima. How do you deal with this ill-posedness?

2. What is $z_t$ and what is $y_t$? Are they different random variables? Based on the paper, it looks like both $y_t$ and $z_t$ denote the latent code, meaning that they are the same variable. I understand that the authors used $y_t$ and $z_t$ to distinguish between the prior and posterior latent code. But the way it is formulated now is not correct. For example, how do you define $KL[q(z)\|p(y)]$ in eq. (3)? Shouldn't It be $KL[q(z)\|p(z)]$ or $KL[q(y)\|p(y)]$?

3. What is the probabilistic model for $p(x_{1:t}|z_{1:t})$? Is it a parametric distribution, e.g., normal distribution? How do you compute the log likelihood function?

4. If the modulus of the eigenvalues of $A$ is not strictly 1, i.e., $|\lambda| =1$, the system either grows or decays exponentially fast. It should be a hard constraint, not a soft constraint. How do you guarantee this?

5. Based on Eq (6), $y_t$ becomes deterministic once $y_{t-1}$ is observed. Then the probability distribution becomes a delta function, $p(y_t|y_{1:t}) = p(y_t|y_{t-1}) = \delta (y_t - Ay_{t-1})$. How do you compute the KL divergence of the Dirac delta distribution?

6. How do you find the correct initial condition $y_0$ to represent $x_{1:T}$. As discussed by the authors, $x_{1:T}$ is transformed to $y_{1:T}$. Then, since the model is linear, once $y_0$ is determined, the rest of the sequence is determined as $y_t = A^t y_0$. Hence, it is crucial to find $y_0$ that describes $x_{1:T}$ the best. How do you choose $y_0$ and how do you guarantee that the choice of $y_0$ is the optimal?

---

> ### Author Response · Authors · 2023-11-17
>
> We would like to thank Reviewer HuLD for identifying the simplicity of our model and our strong results. Furthermore, we appreciate their valuable contributions in posing important questions and seeking clarifications that contribute to the enhancement of our paper. In the following, we provide responses to the comments raised by Reviewer HuLD. Given the chance, we would be pleased to integrate the suggested modifications outlined below into a final revision.
>
> >... their model is, in fact, more similar to the dynamic linear model, or Kalman filter.
>
> We agree with Reviewer HuLD that the dynamic linear model and Kalman filter are related to our approach. The main difference between these techniques and Koopman-based methods is the domain where the dynamics are investigated. For the dynamic linear model and Kalman filter it is the state space, whereas in Koopman methods it is the observable (function) space. In practice, given inputs from the state space $x_{1:T}$, the dynamic linear model and Kalman filter can be viewed as studying a dynamical system $A$ respecting the equation $x_t = A x_{t-1}$. In contrast, Koopman approaches transform states to the observables domain, and study the dynamics there. Namely, $\psi(x_t) = A \psi(x_{t-1})$, where $\psi$ is a nonlinear transformation. Our framework, KVAE, uses nonlinear encoder and decoder, and it employs linearity in the latent space. Therefore, we believe it aligns better and can be justified more profoundly from a Koopman perspective.
>
> >Also, one of the claims is that the model can take care of irregular time series, the capability is simply using a pre-existing model, neural controlled differential equations.
>
> Indeed, we use NCDE to handle irregularly sampled time series information, but we do not state our model's ability to handle irregular time series as a contribution. The ability to handle irregular time series is simply a feature of our model. In retrospect, we believe that utilizing NCDE is a strength of our framework as it provides strong results and it could be potentially replaced by any other module that interpolates time series data.
>
> >... How do you deal with this ill-posedness?
>
> Thank you for raising this great question. In our experience, we did not observe any issues during training or inference due to the mentioned ill-posedness. Furthermore, our ablation study in the revised version Section 5.3 and Table 4 considers a model whose prior is not learned ($\alpha=0$, no GRU). The results show a significant drop in discriminative scores in comparison to KVAE's performance. Thus, while jointly optimizing the prior and posterior may be ill-posed in theory, the strong performance we obtain outweighs this potential shortcoming. If given the opportunity, we would be happy to discuss this limitation in the final version.
>
> >What is $z_t$ and what is $y_t$? Are they different random variables? ...
>
> Regrettably, there seems to be a typo in our notation. Thank you for pointing this out. We fixed this issue in the revised version, where we always use $z$ for the latent variable.
>
> >What is the probabilistic model for $p(x_{1:t} | z_{1:t})$? Is it a parametric distribution, e.g., normal distribution? How do you compute the log likelihood function?
>
> We would like to thank the reviewer for pointing this out. Following a common choice in probabilistic modeling, we assume that $p$ is a Gaussian distribution whose mean is the output of the decoder. Under this assumption, the log likelihood becomes a simple MSE between $x_{1:T}$ and the output of the decoder,
> $\tilde{x}_{1:T}$.
>
> A similar reasoning applies also for the $L_\text{pred}$ term. We added a new discussion in the revised version Appendix C, describing technical details related to the evaluation of the loss function.

---

> > ### Author Response · Authors · 2023-11-17
> >
> > >If the modulus of the eigenvalues of $A$ is not strictly 1, i.e., $|\lambda| = 1$, the system either grows or decays exponentially fast. It should be a hard constraint, not a soft constraint. How do you guarantee this?
> >
> > We constrain the eigenvalues by adding a penalty term $L_\text{eig}$ to the objective function, i.e., it is a soft constraint. In general, one could construct a hard constraint by parametrizing A in the special orthogonal group [1] or by decomposing A to its Koopman representation and limiting the eigenvalues to lie on the unit circle [2]. Furthermore, we would like to note that exponential growth and decay modes are ineffective in our framework since we employ the matrix A to the power of 1. That is, we multiply $\bar{z}_{1:T-1}$ by A only once
> >
> > to obtain $z_{2:T}$, and thus, long-term growth and decay effects do not apply. Please see the definition of $z_t$ given by the end of the Prior modeling paragraph in the revised version.
> >
> >     [1] "Cheap orthogonal constraints in neural networks: A simple parametrization of the orthogonal and unitary group" by Mario Lezcano-Casado, and David Martınez-Rubio.
> >
> >     [2] "Physics-informed probabilistic learning of linear embeddings of nonlinear dynamics with guaranteed stability" by Shaowu Pan, and Karthik Duraisamy.
> >
> > >... How do you compute the KL divergence of the Dirac delta distribution?
> >
> > Thank you for noticing this point. We write the explanation with $z_{1:T}$ due to fixing our typo.
> >
> > The distribution $p(z_{1:T})$ is defined as $p(z_{1:T}) = p(\bar{z}_{1:T}) \prod_t \delta(z_t - A \bar{z}_{t-1})$.
> >
> > In practice, the KL divergence for $p(z_{1:T})$ is always evaluated with $z_{1:T}$ associated with $\bar{z}_{1:T}$.
> >
> > Therefore, we approximate $\delta(z_t - A \bar{z}_{t-1})$ with $1$,
> >
> > leading to $p(z_{1:T}) \approx p(\bar{z}_{1:T})$.
> >
> > The KL divergence can be computed for $p(\bar{z}_{1:T})$ using a closed-form expression given its mean and variance (as well as the mean and variance of $q$).
> >
> > We added this discussion to Appendix C in the revised version.
> >
> > >... How do you choose $y_0$ and how do you guarantee that the choice of $y_0$ is the optimal?
> >
> > We would like to clarify a potential misunderstanding.
> >
> > While Eq. (6) specifies a fully linear dynamical system, our approach implements it in an approximative fashion.
> >
> > Specifically, we define z_t = A \bar{z}_{t-1}
> >
> > (after switching $y_t \rightarrow z_t$), and thus, $z_t$ depends on
> >
> > $\bar{z}_{t-1}$ and not on $z_1$. In practice, we take $z_1 = \bar{z}_1$.

---

> > ### Comment · Reviewer_HuLD · 2023-11-20
> >
> > I am not fully convinced with the argument on the difference between the Kalman filters and the present model. Yes. The authors are correct that in the classical Kalman filters everything is linearized to make the training analytically tractable. But recent developments of Kalman filters, for example see Bezenac et al. "Normalizing Kalman Filters for Multivariate Time Series Analysis" (Neurips 2020), nonlinear encoder and decoder are used for the forward and backward projections and the time evolution of the latent state is given by a linear map. The current method seems to be a restrictive case of the Normalizing KF model, as their model explicitly treats $z_t$ as a random variable and allows a time-depedent linear operator. Please, note that I am not trying to belittle the present study. But I think it is worth mentioning the similarity between the current model and the well-established classical approach.

---

> ### Comment · Reviewer_HuLD · 2023-11-20
>
> Now, I am confused.
>
> 1.  is $p(z_{1:T}) = p(\bar{z}_{1:T}) \prod_t \delta (z_t - A \bar{z}_t-1)$ a proper distribution? Does it normalize to one? Why does $\bar{z}$ not appear in the left-hand side? I don't understand how you can approximate a Dirac delta function by one.
>
> 2. I don't understand why the eigenvalues of $A$ is not so important in this method. As the authors wrote, they may compute $z_{2:t}$ by multiplying $A$ to $\bar{z}$. But this is just a way to construct a linear system. At the end, since they are using the Koopman operator, the latent state should satisfy the relation, $z_{t+n} = A^nz_t$. Then, they end with the same problem. If one of the eigenvalues is bigger than one, $z_{t+n} \rightarrow \infty$, or if less than zero, the model end up with a degenerate system.

---

> > ### Author Response · Authors · 2023-11-21
> >
> > >I don't understand why the eigenvalues of $A$ is not so important in this method. ...
> >
> > The eigenvalues of $A$ are important to the extent of a single application of $A$ to $\bar{z}$. Please note that $z\_t$ is dependent only on $\bar{z}$ variables, and in particular, $z\_t$ is **independent** of $z$ variables. Formally, we define $z_t := A \bar{z}\_{t-1}$. The matrix $A$ is constructed by solving a linear system defined from $\bar{z}$ variables. Specifically, we construct two matrices $\bar{Z}\_0$ and $\bar{Z}$ composed of $\bar{z}\_{0:T-1}$ and $\bar{z}\_{1:T}$ in their columns, respectively. $A$ is the solution to the system $A \bar{Z}\_0 = \bar{Z}$. Finally, $\bar{z}$ variables are sampled from the prior distribution. Therefore, the relation you mentioned, $z\_{t+n} = A^n z\_t$ is **not** enforced explicitly. This is true for any $n$, even for $n=1$. Moreover, our code does not compute powers of $A$. Further, we believe that $|z\_{t+n} - A^n z_{t}|\_2^2$ is expected to grow as $n$ grows.
> >
> > Here is a snippet from our code, responsible for the above computations:
> > ```
> > # zbar is in batch x sequence x features (b x T x k)
> > k = zbar.shape[-1]
> > zbar_p, zbar_f = zbar[:, :-1], zbar[:, 1:]
> > A = torch.linalg.pinv(zbar_p.reshape(-1, k)) @ zbar_f.reshape(-1, k)
> > z_f = zbar @ A
> > z = torch.cat((zbar[:, 0].unsqueeze(1), z_f[:, :-1]), dim=1)
> >
> > pred_loss = F.mse_loss(z, zbar)
> > ```
> >
> > We hope that this discussion clarifies this point. We are happy to make it clearer if needed.

---

> > ### Author Response · Authors · 2023-11-21
> >
> > >is $p(z\_{1:T}) = p(\bar{z}\_{1:T}) \prod\_t \delta(z\_t - A\bar{z}\_{t-1})$ a proper distribution? Does it normalize to one? Why does $\bar{z}$ not appear in the left-hand side? I don't understand how you can approximate a Dirac delta function by one.
> >
> > We are happy to clarify this point. The distribution $p(z\_{1:T})$ is defined as follows
> >     $$p(z\_{1:T}) := \int\_{\bar{z}} p(z\_{1:T}, \bar{z}\_{1:T}) d \bar{z} =
> >     \int\_{\bar{z}} p(z\_{1:T} | \bar{z}\_{1:T}) p(\bar{z}\_{1:T}) d \bar{z} = \int\_{\bar{z}} p(\bar{z}\_{1:T}) \prod\_t \delta(z\_t - A \bar{z}\_{t-1}) d \bar{z}.$$
> > It is a proper distribution under the generalization of distributions that include a Dirac delta distribution. In practice, we evaluate $p(z\_{1:T})$ by summing over sampled batches of $\bar{z}$ instead of integrating. In addition, we replace the Dirac delta with the Kronecker delta function. Thus, our discrete version of $p(z\_{1:T})$ reads
> > $$p(z\_{1:T}) = \sum\_{\bar{z}'} \delta(\bar{z}\_{1:T}-\bar{z}\_{1:T}') p(\bar{z}\_{1:T}) \prod\_t \delta(z\_t - A \bar{z}\_{t-1}),$$
> > where $\delta$ is the Kronecker delta. Finally, we approximate the above discrete distribution by setting $\delta(z\_t - A \bar{z}\_{t-1})$ to $1$, since all $z_t$ that we use in practice are associated with a corresponding $\bar{z}\_{t-1}$. We will improve the discussion in the appendix with the above details.
> >
> > We want to thank the reviewer for furthering the discussion on this point. We are committed to clarifying any remaining concerns the reviewer may have.

---

> ### Author Response · Authors · 2023-11-21
>
> >I am not fully convinced with the argument on the difference between the Kalman filters ... But I think it is worth mentioning the similarity between the current model and the well-established classical approach.
>
> Thank you for sharing this reference. We totally agree with the reviewer that our approach should be also discussed and related to the Kalman filter and the above mentioned work. In the final version, we will cite this work and discuss it in the background section.

---

### Official Review · Reviewer_SWgv · 2023-10-31

**Soundness:** 3 good
**Presentation:** 3 good
**Contribution:** 3 good
**Rating:** 6
**Confidence:** 4

**Summary:**

A variant of VAE for time-series data is proposed. Technically a notable point of the proposed model lies in its prior model. It first samples a sequence $\bar{y}$ based on the outputs of a GRU, thus the dynamics of $\bar{y}$ can be nonlinear. Each $\bar{y}$ is refined to be $y$ by the linear transformation with the DMD matrix computed on the sequence of $\bar{y}$. Then there appears a regularization term to minimize the discrepancy between $y$ and $\bar{y}$, which effectively imposes "soft" linearity on the dynamics of $y$, the final output of the prior model.

**Strengths:**

The proposed method is reasonable, and the experiments are convincing enough to see the superiority of the method especially in terms of generation.

The literature is nicely reviewed, and the paper is adequately placed in the relevant contexts.

I cannot really assess the novelty and the significance in terms of time-series generation. On the other hand, in terms of Koopman-operator-based neural net architectures, the proposed model seems somewhat novel yet technically straightforward.

**Weaknesses:**

From a purely technical point of view, the contribution might look rather incremental. So the paper should be assessed rather in the context of time-series generation models, on which I am not really an expert and thus cannot provide an accurate evaluation.

There is a GRU in the decoder part, which makes it a little difficult to assess the benefit of the Koopman-based prior model. As GRU can provide a nonlinear sequence-to-sequence transformation, it is unclear if the linear structure of $y_{1:T}$ was really beneficial when generating $x_{1:T}$. The results could be more convincing if the decoder did not have the GRU; instead, it should have had a nonlinear **pointwise** (i.e., not sequence-to-sequence) transformation such as a multilayer perceptron applied to each timestep independently. An ablation study with such a change of architecture would be highly informative.

----

Below are minor points.

- Why do you use two different letters, $y$ and $z$, for the prior part and the posterior part, respectively? Usually in VAE papers, the latent variable is always $z$, and we just say $p(\cdot)$ for prior and $q(\cdot)$ for posterior. The current notation in the paper might also be okay, but I just wondered if there could be particular intention to use the two letters.
- Although the paper focuses on the generation capability of the models, some more experiments on the reconstruction / inference capablity could also be interesting.

**Questions:**

(1) As stated above, the presence of GRU in the decoder makes it a little difficult to assess the real utility of the linear structure in the prior model. Do you have some observations when you did not use a nonlinear sequence-to-sequence model in the decoder?

(2) In practice, how linear the sequence of $y$ is? In my understanding, the linearity of the dynamics of $y_{1:T}$ is not a hard constraint but rather is imposed in a soft manner as regularization. I am curious to what extent the $y$ could become linear with such a soft constraint.

---

> ### Author Response · Authors · 2023-11-17
>
> We would like to thank Reviewer SWgv for acknowledging the novelty of our approach, the positioning of the paper, and the compelling experiments. We also would like to thank them for their detailed comments, questions and suggestions that help to deepen our discussion and improve the paper. Below, we address the comments raised by Reviewer SWgv. Given the opportunity, we will be happy to incorporate the modifications listed below into a final revision.
>
> >... An ablation study with such a change of architecture would be highly informative.
>
> To assess the contribution of our Koopman-based prior, we suggested the ablation study in Table 4. There (after fixing the typo $\beta \rightarrow \alpha$), we tested the behavior of the model without the predictive loss term, i.e., no linearity is imposed. Our results demonstrate that KVAE performs better in the majority of cases in comparison to the model with $\alpha=0$. Thus, while the GRU decoder may help in obtaining good results, it is not enough, and the Koopman-based prior improves these results further.
>
> When designing KVAE, we opted for a GRU decoder in order to match existing generative models for a fair comparison. Modifying the decoder will have a strong impact on the performance. Naturally, using a stronger decoder may improve the results significantly, giving an unfair advantage to the approach. Similarly, employing a weaker decoder may damage the results. Thus, the decoder in all existing baselines should change to an MLP decoder for a fair and consistent comparison.
>
> >Why do you use two different letters, $y$ and $z$, for the prior part and the posterior part, respectively? ...
>
> Regrettably, there seems to be a typo in our notation. Thank you for pointing this out. We fixed this issue in the revised version, where we always use $z$ for the latent variable.
>
> >Although the paper focuses on the generation capability of the models, some more experiments on the reconstruction / inference capablity could also be interesting.
>
> Thank you for suggesting that. Following your suggestion, we added two new figures to the revised version showing several reconstruction and inference examples. Specifically, we plot in Fig. 15, the reconstructed signals in the regular setting for all datasets (dashed lines) vs. the ground-truth data (solid lines). In addition, we show in Fig. 16, the inferred signals (dashed lines) in the irregular 50\% case. This experiment demonstrates the inference capabilities of our model since half of the samples were removed during training. We plot the complete signals (solid lines) for comparison.  In addition, we want to emphasize that our climate experiments could also be considered as inference test cases. To evaluate our model abilities in the climate setting, we omit some of the signals, which we refer to as "points of interest", and generate them to compare with the ground truth.
>
> >... Do you have some observations when you did not use a nonlinear sequence-to-sequence model in the decoder?
>
> In general, the predictive loss term improves the results in comparison to models without the predictive loss term (i.e., $\alpha=0$). This is shown in our ablation study in Table 4. In addition, our preliminary results with an MLP decoder show a similar trend. For further details regarding using an MLP decoder, please see the response above.
>
> >In practice, how linear the sequence of $y$ is? ...
>
> Indeed, we preferred to promote linearity using a soft constraint instead of a hard constraint. We believe that it provides the learning algorithm with the flexibility to enforce linearity if possible. In addition, if the data is too complex or the encoder is not expressive enough, the predictive loss term will be high, serving as a tool for identifying potential modeling problems. In practice, the datasets we consider in this work admit a high-level of learnable linearity as we show in Table 1 below. Specifically, we report the relative predictive loss error, i.e.,
> $$|z_{2:T} - A\bar{z}_{1:T-1}|^2 / |\bar{z}_{1:T-1}|^2$$,
> where $z_{2:T}, \bar{z}_{1:T-1}$ are the latent variables and $A$ is the approximated Koopman operator. Importantly, all errors are below $6e-2$, where some errors are one- and two-orders of magnitude below that.
>
> | Data   |   | Regular   | 30\%      | 50\%      | 70\%      |
> |--------|---|-----------|-----------|-----------|-----------|
> | Sines  |   | $4.14e-3$ | $2.32e-3$ | $4.29-3$  | $1.32e-2$ |
> | Stocks |   | $1.11e-3$ | $4.84e-2$ | $5.86e-2$ | $3.35e-4$ |
> | Energy |   | $5.79e-4$ | $3.88e-4$ | $2.23e-3$ | $6.73e-3$ |
> | MuJoCo |   | $1.81e-2$ | $9.22e-3$ | $1.35e-4$ | $8.20e-3$ |

---

> > ### Comment · Reviewer_SWgv · 2023-11-21
> >
> > Thank you for the response. After revisiting the paper, I would like to say that the idea of the soft constraint of linearity of latent variables sounds great. And the additional results provided about the attained linearity can also be a nice piece of information as experimental results if elaborated more.
> >
> > By the way, on the decoder:
> >
> > > When designing KVAE, we opted for a GRU decoder in order to match existing generative models for a fair comparison. Modifying the decoder will have a strong impact on the performance. Naturally, using a stronger decoder may improve the results significantly, giving an unfair advantage to the approach. Similarly, employing a weaker decoder may damage the results. Thus, the decoder in all existing baselines should change to an MLP decoder for a fair and consistent comparison.
> >
> > Although I understand the practical issue of experimentation as the authors pointed out, I still think that the use of a nonlinear RNN in the decoder of "Koopman (V)AE" is somewhat misleading because we usually premise that the state, $x_t$, can be hopefully recovered from the value of the observable, $f(x_t)$. When the decoder is an RNN, the view becomes different because an RNN may use the information of $f(x_t')$ where $t' \neq t$ to reconstruct $x_t$. This is why I commented about the use of GRU in the decoder.
> >
> > However on second thought, I now think that the use of RNN as decoder may be somewhat justified by considering that $x_t$ cannot be recovered only from $f(x_t)$.
> >
> > On the other hand, the use of RNN in the encoder side is easier to justify because it can be regarded as a kind of nonlinear delay-coordinate embedding.

---

> > > ### Author Response · Authors · 2023-11-21
> > >
> > > Thank you for your additional feedback. We are happy to include the attained linearity results in the final version, and to extend the discussion on this topic.
> > >
> > > We believe our view on the encoder and decoder aligns well with many of the reviewer's comments. Specifically, classical Koopman formalism assumes autonomous dynamical systems, i.e., $x\_{t+1}$ can be fully determined from $x\_t$ by the dynamical system update rule $\varphi$. However, this is not necessarily the case for many types of systems, including the dynamics we consider in this work. In this context, the encoder and decoder serve as finding a representation $x\_t'$ that is autonomous, as well as finding the Koopman observable associated with $x\_t'$, denoted by $f\_t'$. Namely, if the encoder is denoted by $\psi$, then our view is that $\psi = \psi\_\text{obs} \circ \psi\_\text{auto}$, where $\psi\_\text{obs}$ learns the latent observable representation, and $\psi\_\text{auto}$ learns the (latent) autonomous state representation. A similar construction and justification can be made for the decoder. In practice, we do not separate $\psi$ into two different encoders. We are happy to extend our discussion on Koopman with respect to this view in the final version.

---

### Official Review · Reviewer_kti2 · 2023-10-31

**Soundness:** 3 good
**Presentation:** 3 good
**Contribution:** 3 good
**Rating:** 6
**Confidence:** 4

**Summary:**

This paper proposes a variational autoencoder, Koopman VAE (KVAE), for time series data based on Koopman theory. The idea is to use a linear map to represent the prior dynamics, alongside a nonlinear coordinate transformation (the encoder) that maps the data to a linear representation. The main features of KVAE is that (i) it can incorporate domain knowledge (in the prior) by placing constraints on the eigenvalues of the linear map; (ii) the behaviour of the system can be analysed using dynamical systems theory tools. The results in the paper are promising, showing that KVAE outperforms SOTA GANs and VAEs across synthetic and real world time series generation benchmarks.

**Strengths:**

- Experimental results indicate strong performance compared to GANs and VAEs
- The use of linear latent dynamics simplifies the learning of the latent dynamics and allows for adding physical constraints, as indicated in section 4.3

**Weaknesses:**

- A large literature of sequential VAEs for time series data generation is omitted e.g. [1,2,3,4], despite a large number of baselines being used in the experiments section. Considering there is heavy development in this area, it would be useful to compare KVAE to these methods.
- More discussion in the experiments section is required on the topic of analysing "the behaviour of the system...using dynamical systems theory tools" in order to claim this as an additional feature of KVAE


[1] Chung et al. (2015). A Recurrent Latent Variable Model for Sequential Data

[2] Rubanova et al. (2019). Latent ODEs for Irregularly-Sampled Time Series

[3] Li et al. (2020). Scalable Gradients for Stochastic Differential Equations

[4] Zhu et al. (2023). Markovian Gaussian Process Variational Autoencoders

**Questions:**

N/A

---

> ### Author Response · Authors · 2023-11-17
>
> We extend our appreciation to Reviewer kti2 for recognizing the robust performance of our approach, its simplicity, and its capability to incorporate physical constraints. Moreover, we are grateful for their insightful comments and suggestions aimed at enhancing the paper and delving into a more profound discussion. Below, we provide responses to the comments raised by Reviewer kti2. If given the opportunity, we would gladly integrate the suggested modifications outlined below into the final revision.
>
> >A large literature of sequential VAEs for time series data generation is omitted e.g. [1,2,3,4], despite a large number of baselines being used in the experiments section. Considering there is heavy development in this area, it would be useful to compare KVAE to these methods.
>
> Thank you for bringing these works to our attention. We agree with the reviewer that sequential VAEs are actively developed in recent years, on a broad range of applications. In our work, we focused on comparing with baseline generative approaches that target the same benchmarks and evaluation environment, for a fair and consistent evaluation of approaches. That being said, we are also happy to consider papers [1-4]. Specifically, paper [1] can be viewed as a close variant of our approach. In particular, omitting the predictive loss term in KVAE forms method [1]. The results of this method appear in our ablation study in Table 4 in the submitted paper. Please notice that there was a typo there, and all appearances of $\beta$ should be replaced with $\alpha$ (we have corrected this in the revised manuscript). Further, we are currently running method [2], and we will report the results as soon as we have them. As for method [3], it can be viewed as a VAE only in the time-continuous limit. Nevertheless, we are currently working on migrating the available code to fit into our training and evaluation framework. Finally, we could not find an open-source implementation of method [4], unfortunately. We started to implement this approach from scratch, however, the time constraints of the rebuttal may be too limiting to obtain preliminary results.
>
> >More discussion in the experiments section is required on the topic of analysing "the behaviour of the system...using dynamical systems theory tools" in order to claim this as an additional feature of KVAE
>
> Following the reviewer's suggestion, we extended the discussion in Section 5.2 on the physics-constrained generation example. The revised text details the particular loss term $\mathcal{L}_\text{eig}$ that we use. Moreover, we analyze the nonlinear pendulum system from a stability viewpoint, where Koopman eigenvalues represent growth and decay modes. The discussion is complemented by Figure 3, which shows the spectra of Koopman operators for unconstrained KVAE vs. constrained KVAE. Our results indicate that constraining the eigenvalue modulus improves generation results and positively affects the operator's spectrum. We also compute the spectra associated with the computed operators in the regular setting and further discuss the results in Appendix F.

---

> > ### Author Response · Authors · 2023-11-21
> >
> > In the table below, we report the discriminative results obtained with LatentODE [2] and LatentSDE [3]. Overall, LatentODE performs similarly to TimeGAN. The results for LatentSDE are extremely poor, however. There could be several reasons for this observed behavior: i) the paper assumes a certain SDE model which may not apply to the datasets we consider; ii) the examples considered in [3] are low-dimensional in terms of features. Specifically, we followed the example in Sec. 7.2 (Lorenz). It may be that the authors' code does not generalize to data with multiple features; iii) in our experiments, the hyperparameters of LatentSDE are tuned to match the memory footprint used in our benchmarks by all methods. It may be that LatentSDE needs more parameters to solve the tasks in the benchmarks. We now run LatentSDE with extended hyperparameters, however, even if the results improve, this will not be a fair comparison.
> >
> > | Method        | Sines   | Stocks  | Energy  | MuJoCo  |
> > |---------------|---------|---------|---------|---------|
> > | LatentODE [2] | $0.013$ | $0.111$ | $0.496$ | $0.178$ |
> > | LatentSDE [3] | $0.480$ | $0.471$ | $0.500$ | $0.487$ |
> > | TimeGAN       | $0.011$ | $0.102$ | $0.236$ | $0.409$ |
> > | Ours          | **$0.005$** | **$0.009$** | **$0.143$** | **$0.076$** |

---

### Meta-Review · Area_Chair_Ptwo · 2023-12-05

**Metareview:**

This paper proposed a new sequential VAE model and the main difference from the existing ones is in the temporal prior definition. The temporal prior is hierarchical, first uses a GRU to get the top-layer latent states, and use them for estimating the best linear dynamical system fitting, then use the estimated linear dynamical system to generate the lower-layer latents. Experiments compared the model with some GAN and VAE baselines on time-series, showing substantial improvements.

Reviewers have questions regarding presentation clarity as well as the missing literature on recent progress of sequential VAE, which is largely addressed in revision.

Just one small point: there is another paper that proposed Kalman VAE and used the shorthand name KVAE.

https://sites.google.com/view/kvae

**Justification For Why Not Higher Score:**

Still need justifications in terms of high novelty regarding sequential VAE literature.

**Justification For Why Not Lower Score:**

Presented an interesting way of constructing temporal prior for sequential VAE that the community would be happy to see.

---

### Decision · Program_Chairs · 2024-01-16

Accept (poster)